# Structure-guided discovery of Otopetrin 1 inhibitors reveals druggable binding sites at the intrasubunit interface

Batuujin Burendei [1,4], Joshua P. Kaplan [2,3,4], Gerardo M. Orellana [2], Emily R. Liman [2,3,5] ✉, Stefano Forli [1,5] ✉ & Andrew B. Ward [1,5] ✉

Proton conductance across cell membranes serves many biological functions, ranging from the regulation of intracellular and extracellular pH to the generation of electrical signals that lead to sour taste perception. Otopetrins (OTOPs) are a conserved, eukaryotic family of proton-selective ion channels, one of which (OTOP1) serves as a gustatory sensor for sour tastes and ammonium chloride. As the functional properties and structures of OTOP channels were only recently described, there are presently few tools available to modulate their activity. Here, we perform subsequent rounds of molecular docking-based virtual screening against the structure of zebrafish OTOP1, followed by functional testing using whole-cell patch-clamp electrophysiology, and identify several small molecule inhibitors that are effective in the low-to-mid μM range. Cryo-electron microscopy structures reveal inhibitor binding sites in the intrasubunit interface that are validated by functional testing of mutant channels. Our findings reveal pockets that can be targeted for small molecule discovery to develop modulators for Otopetrins. Such modulators can serve as useful toolkit molecules for future investigations of structure-function relationships or physiological roles of Otopetrins.

Proton conductance is involved in many physiological processes, ranging from the regulation of intracellular and extracellular pH to the generation of action potentials in dinoflagellates and the initiation of sour taste signaling in vertebrates and invertebrates[1,2]. While there are many plasmalemmal ion channels and transporters that allow for proton conductance, selectivity for protons was previously observed only in the case of the voltage-gated proton channel $H_V1$[3,4] and influenza proton channel M2[5]. In 2018, we reported the identification of a family of proton-selective ion channels, collectively called the Otopetrins or OTOPs[6]. The OTOP proteins are evolutionarily conserved from nematodes to humans[7] and unrelated in sequence or structure to known transporters or ion channels[8,9].

All members of the family of OTOP proton channels studied thus far, including vertebrate OTOP1, OTOP2, OTOP3, and various invertebrate OTOPs, such as *Caenorhabditis elegans* and *Drosophila* Otop-like proteins, function as inward conducting, voltage-insensitive proton channels, with variation in channel properties observed among orthologs and paralogs[2,6,10,11]. OTOP1 was first identified based on its essential function in the development of calcium carbonate-based otoconia in the vestibular system of mice[12] and otoliths in zebrafish[13]. More recently, OTOP1 was shown to be expressed in vertebrate taste receptor cells, where it functions as a receptor for sour and ammonium tastes[14–17], while a related protein (Otopetrin-like a) is required for gustatory responses to acids in invertebrates[18,19]. OTOP1 is also expressed in various other tissues in mammals, such as the heart,

[1]Department of Integrative Structural and Computational Biology, The Scripps Research Institute, La Jolla, CA, USA. [2]Section of Neurobiology, Department of Biological Sciences, University of Southern California, Los Angeles, CA, USA. [3]Program in Neuroscience, University of Southern California, Los Angeles, CA, USA. [4]These authors contributed equally: Batuujin Burendei, Joshua P. Kaplan. [5]These authors jointly supervised this work: Emily R. Liman, Stefano Forli, Andrew B. Ward. ✉e-mail: liman@usc.edu; forli@scripps.edu; andrew@scripps.edu

uterus, adrenal gland, mast cells, and adipose tissue, where its physiological roles remain poorly understood[6,20,21].

OTOP channels are highly and possibly perfectly selective for protons, with most variants exhibiting inward currents in response to low extracellular pH, while some also efflux protons in response to extracellular alkalinization[6,22,23]. Protons are not just the permeant ion but also gate some OTOP channels. We recently showed that OTOP1 and OTOP3, but not OTOP2, are proton-gated through a mechanism involving multiple extracellular loops[22], and another study identified a histidine on one of them as a key proton sensor[24]. OTOP1 and OTOP3 are also gated by the trace metal ion zinc ($Zn^{2+}$), which also inhibits most OTOP channels[6,25,26], through interactions with key residues on extracellular linkers L5-6 and L11-12, which connect transmembrane helix 5 (TM5) and TM6, and TM11 and TM12, respectively[25]. More recently, OTOP1 was shown to be gated in response to ammonium chloride, an effect that requires a highly-conserved intracellular arginine residue[16]. In addition to $Zn^{2+}$, Cibacron Blue 3GA[27] was recently shown to inhibit OTOP1, and suramin was shown to inhibit an OTOP1-dependent $Ca^{2+}$ response[28]. Notably, these are promiscuous molecules known to modulate other ion channels[29–33] and the mechanisms by which they inhibit OTOP channels are not known. Historically, studies of ion channels have greatly benefited from the use of a toolkit of small molecules in the form of agonists or antagonists that can be used to identify key structural elements, modulate channel activity, and assign contributions to specific channels in complex physiological contexts[34–39]. Recently, three positive modulators of mouse OTOP1 were reported[40], but as of now, no specific inhibitors of any OTOP channel have been identified. Such molecules would be useful to identify physiological roles of OTOP channels in non-model systems where knockout animals are unavailable, and could be used therapeutically, in cases where overactive OTOP channels could have detrimental effects.

Here, we use molecular docking and electrophysiology to virtually screen a library of drug-like small molecules against zebrafish (*Danio rerio*) OTOP1 (DrOTOP1) and validate them as functional channel modulators. We identify several inhibitors from the first round of screening and improve our results through another round of screening. Cryo-electron microscopy (cryo-EM) structure determination and complementary mutagenesis allow us to identify the binding sites for these compounds, which represent pockets that can be targeted for future drug discovery.

## Results

### Virtual screening identifies potential OTOP1 inhibitors

Previously, cryo-EM structures of DrOTOP1, chicken (*Gallus gallus*) OTOP3, *Xenopus tropicalis* OTOP3, *C. elegans* OTOP8 (CeOTOP8), and mouse (*Mus musculus*) OTOP2 were determined, revealing that Otopetrins form dimers, with each monomer consisting of twelve transmembrane helices divided into N-domain and C-domain halves that adopt pseudosymmetric barrel-shaped folds[8,9,41] (Fig. 1A). The structures revealed a central lipid-filled tunnel unlikely to support ion permeation[8] (Fig. 1A, black oval). All-atom molecular dynamics simulations revealed three locations within each monomer where water wires could form, within the N-domain, C-domain, and the intrasubunit interface, each of which could act as a conduit for proton conductance[8] (Fig. 1A, red circles). The dimerization of OTOP channels is required for cell surface trafficking[8] and recent work on CeOTOP8 suggests that movement of the subunits relative to each other may play a role in channel gating[41]. Because there is no central permeation pathway, we treated each monomer of the OTOP dimer as an independent proton-conducting unit. We focused on the putative pores as regions of interest for virtual screening, as we hypothesized that disrupting the water wires therein would affect channel activity.

To identify suitable binding pockets for small molecules, we analyzed a monomer of the DrOTOP1 cryo-EM model (PDB: 6NF4) determined at pH 8.0, which is expected to correspond to a closed state of the channel, using AutoSite[42] to identify potential ligand-binding sites. Among these predictions, we identified suitable pockets in each region of interest, the N-domain pocket, the intrasubunit interface pocket, and the C-domain pocket (Fig. 1B). From here, we prioritized the C-domain pocket because it was more buried, had more polar residues available for protein-ligand hydrogen bonds (H-bonds), and the C-domain putative pore was previously shown to harbor a salt-bridge between E429 - R572 that is required for channel function[8] (Fig. 1C). Thus, by targeting a closed state of the channel at a pocket near one of the putative pores, we expected to identify molecules that could inhibit channel activity.

Supplementary Fig. 1 summarizes our virtual screening approach. We used AutoDock Vina v1.1.2[43] to dock a 90% diversity set consisting of 302,893 molecules from the ChemBridge Library of drug-like molecules[44], using a docking box that fully encapsulated the C-domain pocket. Docking results were filtered with hard cutoffs of predicted free energy of binding (docking score), ligand efficiency, and number

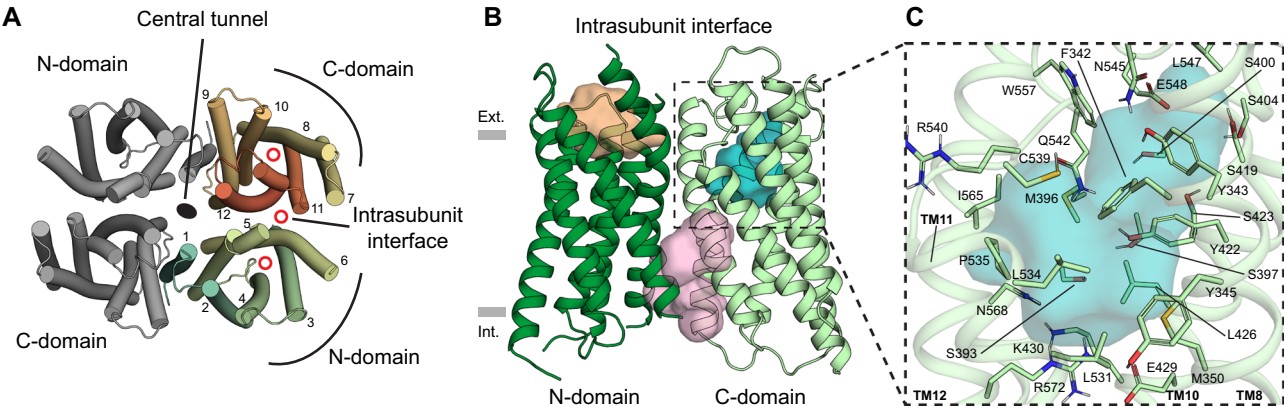

**Fig. 1 | Pocket identification for virtual screening on the zebrafish OTOP1 cryo-EM structure. A** Top view of the zebrafish OTOP1 (DrOTOP1) dimer (PDB: 6NF4) with transmembrane helices (TM) shown as cylinders. One monomer is colored in a rainbow spectrum, TM helices 1 to 12 are labeled, and the other monomer is colored gray. Putative pores in the N-domain, C-domain, and the intrasubunit interface are indicated with red circles and the central tunnel between the dimer is indicated with a black oval. **B** Side view of DrOTOP1 monomer depicted as a cartoon, with the N-domain colored dark green and the C-domain colored light green. AutoSite-predicted pockets are shown along the putative pores in the N-domain (orange), C-domain (blue) and the intrasubunit interface (pink). Gray rectangles indicate the height of the lipid bilayer, with the extracellular side (Ext.) on the top and the intracellular side (Int.) on the bottom. **C** Inset of (**B**) showing a zoom-in of the C-domain pocket shown as a transparent surface (blue), with pocket-lining residues labeled and shown as sticks, and the backbone shown as a transparent cartoon.

of predicted H-bonds. As the pocket is small and buried, smaller molecules weighing 220 - 350 Da were favored, which limited the diversity of molecules that passed the filters. Thus, to preserve as broad of a chemical space as possible, we clustered the molecules based on the Tanimoto similarity of their molecular fingerprints and selected molecules through visual inspection focusing primarily on desirable protein-ligand interactions, proper structural conformation of ligands, and consideration of desolvation cost, with the goal of obtaining a final set of chemically diverse molecules. From this analysis, 50 small molecules were selected for functional testing.

## C11 blocks OTOP1 inward conducting current in whole-cell patch-clamp

The 50 molecules identified were individually tested by whole-cell patch-clamp electrophysiology for effects on DrOTOP1 expressed in human embryonic kidney (HEK)−293 cells. Each compound was tested at a concentration of 200 μM to allow us to detect a low-affinity inhibitor, anticipating the need for further rounds of virtual screening to identify more potent compounds. DrOTOP1 currents were evoked using an extracellular pH 5.5 $Na^+$-free (NMDG-based) solution for 8 seconds, followed by the application of each compound for 16 s at the same pH, and then an 8 s wash-off, also at the same pH. This approach enabled us to observe the on- and off-responses of each compound. To quantify the effect of each compound, we compared the current magnitude during compound application immediately before wash-off, to the peak inward current following its removal (Supplementary Fig. 2A). We note that this method is expected to reduce identification of false positives but might miss some real hits where the compounds have slow kinetics; such a compound was identified instead by visual inspection (see below). Vehicle control recordings using the highest dimethyl sulfoxide (DMSO) concentration used for any of the compounds (0.3%) showed no measurable inhibition of DrOTOP1 currents (Supplementary Fig. 2A, B). Using this method, we identified five small molecules that inhibited DrOTOP1 currents by over 25% (Fig. 2A, red and gray bars; Supplementary Fig. 3A, B). Notably, compound 11 (C11) at 200 μM nearly fully inhibited the currents (Fig. 2A). Inhibition of DrOTOP1 currents by C11 was dose-dependent, with an $IC_{50}$ of 76 μM and a Hill coefficient of $h = 2.2$, suggesting multiple binding sites and positive cooperativity (Fig. 2B, C).

## Substructure filtering identifies more potent OTOP1 inhibitors

To identify higher potency inhibitors, we used a ligand-based analog-by-catalog search. We focused on C11 and performed a substructure search using different SMARTS[45] patterns describing key structural features that directly engaged protein residues, aiming to find an analog with higher affinity to DrOTOP1 (Supplementary Fig. 3C). This allowed us to identify compounds with substructures matching C11, also known as benzil monoxime (BMO), which consists of two phenyl rings connected by an oxime and a ketone group (Supplementary Fig. 3C, **1**). Specifically, SMARTS **2**–**5** were designed to match molecules with two 5-or-6-membered heteroaromatic rings, connected by common functional groups like amide bonds, esters, ketones, as well as the original oxime and ketone groups of C11 (Supplementary Fig. 3C, **2**–**5**). When inspecting the docked conformation of C11, we noticed that the placement of aromatic rings was reminiscent of the tricyclic dibenzazepine family of Food and Drug Administration (FDA)-approved drugs, such as oxcarbazepine (Supplementary Fig. 3C, **6**). Thus, we also designed SMARTS **7** (Supplementary Fig. 3C, **7**) to filter for molecules with a dibenzazepine-like core but allowing for substitutions of 5 or 6-membered heteroaromatic rings on either side to increase the chemical diversity of the filtered results. The SMARTS **2**–**5** and **7** were used to filter the full ChemBridge library[44], comprising - 1.3 million molecules, using substructure matching in RDKit[46], identifying a final set of 35,908 molecules. This set was used

for a subsequent round of virtual screening, which was performed and analyzed similarly as before (Supplementary Fig. 3D). Once again, through visual inspection of docking results, a final set of 51 molecules was selected for functional testing. In addition, we included three FDA-approved drugs from the dibenzazepine family – oxcarbazepine, eslicarbazepine acetate, and tianeptine.

The second series of compounds was again tested by whole-cell patch-clamp recording of DrOTOP1 expressed in HEK-293 cells. Of the 54 compounds tested, we identified 16 that inhibited DrOTOP1 acid-induced currents by over 25% (Fig. 2D and Supplementary Fig. 3E). The most potent, C2.2, fully blocked currents at 200 μM, and dose-response data were fit with an $IC_{50}$ of 6.6 μM, more than 10-fold higher than the $IC_{50}$ of C11 (Fig. 2E, F and Supplementary Fig. 2B, C). The Hill coefficient of 1.52 indicated positive cooperativity and multiple binding sites, similar to C11. We next tested each of these compounds against mouse OTOP1 (mOTOP1) and human (*Homo sapiens*) OTOP1 (hOTOP1), which are among the closest related proteins in the mammalian proteome. At saturating concentrations (300 μM and 100 μM, respectively), neither C11 nor C2.2 significantly inhibited mOTOP1, while C11 inhibited hOTOP1 currents by - 24% and C2.2 had no effect (Fig. 2G). Thus, both compounds are specific for DrOTOP1 over mammalian orthologs, which is surprising given the sequence similarities present in C-domain pocket residues (Supplementary Fig. 4). However, there is overall low sequence similarity between DrOTOP1 and mammalian OTOP1 (- 40%). For both compounds, we also determined whether there was any pH-dependence to their inhibitory effect by measuring dose-response curves over a range from pH 6.0, where sizable currents first appear, to pH 5.0. Over this range, C11 (Fig. 2H) showed no evidence of pH dependence ($IC_{50}$ values were: pH 6.0: 71.5 μM, pH 5.5: 76.3 μM, and pH 5.0: 75.2 μM). C2.2 (Fig. 2I) was slightly less effective at inhibiting DrOTOP1 currents at pH 5.0, while no difference was observed between pH 6.0 and pH 5.5 ($IC_{50}$ values were: pH 6.0: 5.97 μM, pH 5.5: 6.67 μM, and pH 5.0: 12.6 μM). Overall, neither compound exhibited evidence of clear pH-dependence as observed for inhibition by $Zn^{2+}$ [14,47].

## Effects of C-domain pocket mutations on inhibition by C11 and C2.2

To experimentally verify whether C11 and C2.2 bind in the C-domain pocket as predicted, we used their docked poses to guide mutagenesis. The predicted interactions for C11 and C2.2 include π-stacking with F342 and Y422, potential H-bonding with Q542, N568, and R572 (Fig. 2J, K and Supplementary Fig. 5A), and hydrophobic packing. We generated single-residue mutants of these and other residues in DrOTOP1 that were expected to disrupt specific molecular interactions or to introduce a bulky tryptophan residue (W) to produce steric hindrance. Each mutant was tested using whole-cell patch-clamp electrophysiology to assess its impact on inhibition by C11 at concentrations that were either saturating (300 μM) or near the $IC_{50}$ (100 μM).

Among a broad range of mutations, only one, I565W, significantly altered inhibition by C11, shifting its $IC_{50}$ to 190 μM, a - 2.5-fold increase compared with WT (Supplementary Fig. 5B–D). We also tested a double mutant, I565W / G346F, as G346F slightly reduced inhibition by C11 at 100 μM, but found no further effect as compared with the single mutant (Supplementary Fig. 5B, C). We next tested a subset of these mutants for sensitivity to C2.2, focusing on those that could occlude the C-domain pocket. As with C11, the only mutation that significantly changed sensitivity to C2.2 was I565W, resulting in a - 10-fold shift of the $IC_{50}$ to 82 μM (Supplementary Fig. 5E−H). Two other mutations, Y422F and L531F, showed mildly reduced inhibition by 10 μM C2.2, but did not further reduce sensitivity to C2.2 when mutated in combination with I565W (Supplementary Fig. 5E, F). Thus, despite extensive mutagenesis and testing, summarized in Supplementary Fig. 5I, we could identify only one mutation (I565W),

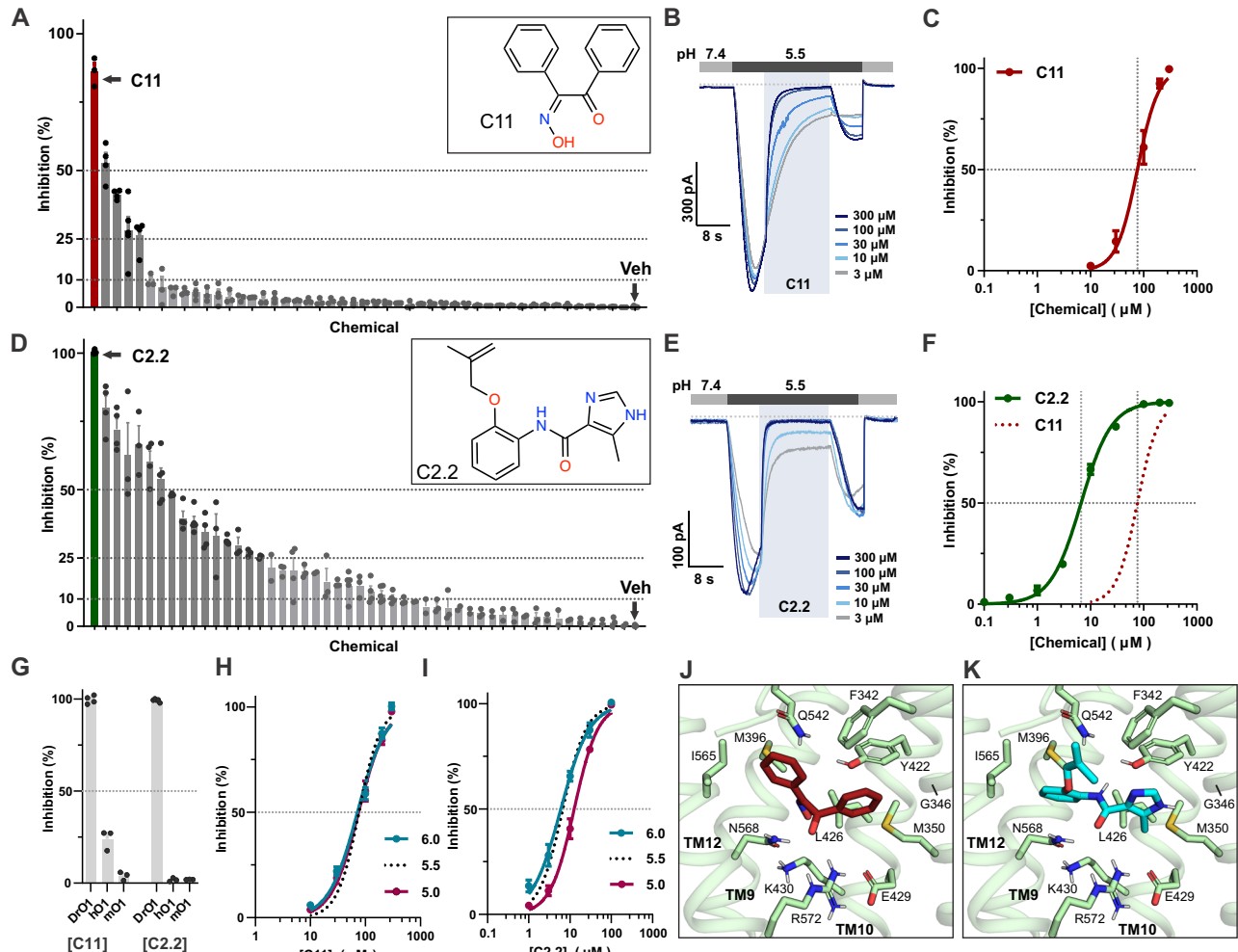

**Fig. 2 | Screening of small molecules against DrOTOP1 identifies inhibitors. A, D** Inhibition of pH 5.5-induced currents by compounds tested (at 200 μM) from screen 1 (**A**) and screen 2 (**D**) measured with whole-cell patch-clamp recording from HEK-293 cells expressing DrOTOP1. $n = 3$–5 cells per compound. C11 (red bar), C2.2 (green bar), and 0.2% DMSO control (Veh, gray bar) are indicated. Compound names are omitted. Bars in dark gray indicate molecules that inhibit DrOTOP1 by > 25%. Insets show chemical structures of C11 and C2.2. **B** Representative traces of HEK-293 cells expressing DrOTOP1 showing dose-dependent inhibition by C11 at pH 5.5. **C** Data from experiments as in (**B**). Data fit with a Hill slope = 2.21 and $IC_{50}$ of 76.3 μM. $n = 4$ cells. **E** Representative traces of HEK-293 cells expressing DrOTOP1 showing dose-dependent inhibition by C2.2 at pH 5.5. **F** Data from experiments as in (**E**). Data fit with a Hill slope = 1.52 and $IC_{50}$ of 6.67 μM. $n = 8$ cells.

**G** Inhibition of DrOTOP1, human OTOP1 (hOTOP1), and mouse OTOP1 (mOTOP1) in HEK-293 cells by C11 and C2.2. $n = 4$ (C11) and 5 (C2.2) cells for DrOTOP1, $n = 3$ cells for hOTOP1, and $n = 3$ cells for mOTOP1. **H, I** Dose-dependent inhibition of C11 (**H**) and C2.2 (**I**) in HEK-293 cells expressing DrOTOP1. Currents were evoked at pH 6.0 and pH 5.0. pH 5.5 data from (**C**, **F**). Data were fit with a Hill slope. C11: pH 6.0 = 71.5 μM, $h = 1.67$ and pH 5.0 = 75.2 μM, $h = 1.88$. $n = 5$ cells (pH 6) and $n = 7$ cells (pH 5). C2.2: pH 6.0 = 5.99 μM, $h = 1.25$ and pH 5.0 = 12.6 μM, $h = 1.53$. $n = 8$ cells (pH 6) and $n = 8$ cells (pH 5). Docked poses for C11 (dark red) (**J**) and C2.2 (cyan) (**K**) in the C-domain pocket. The DrOTOP1 model (light green) is shown as a transparent cartoon and sticks, with pocket-lining residues labeled. All data are presented as mean ± SEM. All experiments performed by holding cells at $V_m = -80$ mV. Source data are provided as a Source Data file.

predicted to generate steric occlusion of the C-domain pocket, that made the channels less sensitive to C11 and C2.2.

## Cryo-EM structures of DrOTOP1 with inhibitors reveal binding pockets

To gain further insight into the interactions of the identified inhibitors with DrOTOP1, we turned to structure determination. Initially, we determined a cryo-EM structure of DrOTOP1 in complex with C11 (DrOTOP1_C11) at 3.7 Å resolution. The structure adopted the same closed conformation as the apo structure (DrOTOP1_apo) (EMDB:9360)[8] (Supplementary Fig. 6A, B), and the resulting model built into the map was highly similar with a root mean square deviation (RMSD) of ~1 Å compared to the DrOTOP1_apo model (PDB:6NF4) (Supplementary Fig. 6C–E). The C-domain pocket region in DrOTOP1_apo was well resolved and free of non-protein

density (Supplementary Fig. 6C). The same region in DrOTOP1_C11 exhibited ambiguous density extending from side chains in the C-domain pocket (Supplementary Fig. 6D, inset), and we were unable to confidently model C11 there (Supplementary Fig. 6F). Overall, we did not observe any conformational changes in side-chains of pocket lining residues, including I565, that could be attributed to binding of the inhibitor (Supplementary Fig. 6G). We concluded that since C11 has an $IC_{50}$ of 76.3 μM (Fig. 2C), its occupancy was likely too low to be determined by cryo-EM. Alternatively, the density could be an artifact caused by DMSO in the sample or noise, as the resolution of DrOTOP1_C11 (3.7 Å) is worse than that of DrOTOP1_apo (3.0 Å). Elsewhere on the structure, we could not identify significant conformational changes or other unassigned density. Thus, neither our structure nor mutagenesis results clearly indicated that C11 binds in the C-domain pocket.

To search for more potential binding sites, we tried again using two more potent inhibitors, C2.2, the most potent inhibitor with an $IC_{50}$ of 6.67 μM, and C2.36, which has a uniquely slow off-rate from the channel (see below). We determined the following structures: DrOTOP1_C2.2 at 3.2 Å resolution (Fig. 3A, B) and DrOTOP1_C2.36 (see below) at 3.4 Å resolution. Comparing DrOTOP1_C2.2 with DrOTOP1_apo, we searched the C-domain pocket for non-protein density, but could not identify any, suggesting that C2.2 was not bound there either (Fig. 3C). Similarly, we observed little conformational change in the sidechains of pocket lining residues, including I565 (Fig. 3D), compared to DrOTOP1_apo. Next, we analyzed the DrOTOP1_C2.2 model for other differences. Upon closer comparison, we found two regions on the intracellular half of the intrasubunit interface that contained more convincing cryo-EM density that could correspond to C2.2 molecules, compared to the C-domain pocket (Supplementary Fig. 7A). We refer to these as the "outer site", facing the lipid bilayer (Fig. 3E, **right**), and the "central site", facing the central cavity (Fig. 3F, **right**). Notably, both sites are exposed to bulk lipid – the outer site to nanodisc lipids and the central site to the lipid-filled central cavity of DrOTOP1 – and were previously identified as cholesterol binding sites in DrOTOP1_apo[8] (Fig. 3E, F **left**). These sites seem evolutionarily conserved, as the residues therein are generally conserved among vertebrate Otopetrins (Supplementary Fig. 7B) – even more so when focusing only on OTOP1 sequences from zebrafish and mammals (Supplementary Fig. 4).

The outer site is flanked by TM3 and TM6 from the N-domain and TM12 from the C-domain (Fig. 3G). We observe local conformational changes such as TM3 shifting inward (Fig. 3G **red arrow**), the sidechains of M133 and M573 splaying away from the site, and a slight rotation in the sidechain of F271 (Fig. 3G). These changes allow the cholesterol binding site to accommodate the C2.2 inhibitor, providing space and ample hydrophobic packing and π-stacking interactions with aromatic residues F271, F570, and W532. The two polar residues in the site, E267 and H574, are highly conserved across all Otopetrins (Supplementary Figs. 4 and 7B), and were previously found to be necessary for channel function[8,9]. Thus, an inhibitor binding nearby may engage their sidechains to disrupt the hydration pattern of the site and impede proton conductance through the channel. Similarly, the central site, located on the opposite side of the monomer, is flanked by TM5 and TM6 from the N-domain, and TM9 from the C-domain (Fig. 3I). Here, we also observed significant shifts in the intracellular ends of TM6, which moves away from the C-domain (Fig. 3I, **red arrow**) and TM9, which moves toward the N-domain (Fig. 3I, **black arrow**). This is likely a concerted movement, as interactions between TM6 and TM9 like the K281 - E384 salt-bridge and hydrophobic packing between V278 and L381 remain present (Fig. 3I). The most pronounced change occurs on TM6, where the sidechain of F277 moves into a space occupied by cholesteryl hemisuccinate (CHS) in DrOTOP1_apo, a cholesterol mimic, to accommodate the binding of C2.2 (Fig. 3I). Based on the shape of the density and accompanying local conformational changes, we conclude that C2.2 binds to both sites and replaces the endogenous cholesterols.

In terms of larger-scale conformational changes, global C-alpha RMSD comparisons of DrOTOP1_C2.2 and DrOTOP1_apo remain below 2 Å, suggesting that the two structures are overall highly similar (Supplementary Fig. 8A). However, per-domain RMSD comparisons revealed that the N-domain harbors more changes than the C-domain (Supplementary Fig. 8A). Visually, we observed TM shifts on the intracellular side of DrOTOP1 (Supplementary Movie 1), where TM3 and TM9 move closer to tighten the intracellular side of intrasubunit interface (Fig. 3H, **red arrows**) and TM6 moves laterally towards the N-domain (Fig. 3H, **black arrow**). Indeed, by color-mapping the C-alpha RMSD values onto DrOTOP1_C2.2, we saw that most of the N- and C-domains remained unchanged and that conformational changes

with RMSD above 2 Å were localized to the intracellular ends of the TMs that flank the outer and central binding sites (Supplementary Fig. 8B). In contrast, the extracellular side of the TM domains exhibit much less movement, with only TM5 shifting slightly from the center (Supplementary Fig. 8B). Thus, the conformational changes resulting from two copies of C2.2 binding simultaneously are highly localized to the binding sites and nearby TMs.

Like DrOTOP1_C2.2, the cryo-EM map of DrOTOP1_C2.36 (Fig. 4A) did not show non-protein density in the C-domain pocket (Fig. 4B), nor were there conformational changes (Fig. 4C). Instead, the map showed occupation with density that corresponds to the C2.36 inhibitor in the outer site (Fig. 4D), highly reminiscent of the density present in DrOTOP1_C2.2 in the same site, as well as very similar conformational changes in residues M133, F271, and M573 (Fig. 4E). The central site also shows density, albeit discontinuous, that could potentially correspond to multiple copies of C2.36 (Fig. 4F). While the density could also correspond to cholesterols or lipids in the central cavity, based on the shifts in F277 and TM6, which are similar to those observed in DrOTOP1_C2.2 (Fig. 4G), we attributed the density to C2.36. Based on the commonalities between the DrOTOP1_C2.36 and DrOTOP1_C2.2 structures, we concluded that inhibitor C2.36 binds the same two intrasubunit sites. Similar to DrOTOP1_C2.2, minimal large-scale conformational changes were observed in the structures of DrOTOP1_C11 and DrOTOP1_C2.36 (Supplementary Fig. 8C, D).

Following the identification of these sites, we used the AutoDock-GPU[48] engine to dock C2.2 and C2.36 into the outer and central sites of their respective cryo-EM models, attempting to reproduce their experimental binding modes in these sites. In the outer site, for both C2.2 and C2.36, docking produces poses that, while not within the conventional RMSD cutoff of < 2 Å, still closely resemble the cryo-EM poses (Supplementary Fig. 9A, B), with docking scores that are within < 1 kcal/mol of their respective best scoring poses from the same docking run (Supplementary Fig. 9C, D). Similarly, for the central site, we obtained poses which more closely resemble the cryo-EM poses (with RMSD < 2 Å) (Supplementary Fig. 9E–H), including for the second copy of C2.36 we modeled (Supplementary Fig. 9I, J). As such, docking produced similar poses that corroborated our cryo-EM modeling of the inhibitors.

## Mutagenesis in outer and central sites confirms binding site of inhibitors

To functionally test the significance of the two binding sites identified by cryo-EM, we generated point mutations of DrOTOP1 using the binding poses and conformational changes observed in DrOTOP1_C2.2 to guide our mutations. As per the binding pose that best fits our cryo-EM density, C2.2 forms the following interactions in the outer site: hydrophobic packing against many non-polar residues W126, G130, M133, L134, F271, F570, and M573, and π-stacking with W532 (Fig. 5A). In addition, C2.2 could potentially form H-bonds with E267, if the sidechain is protonated or a water bridge is present, and with H574 if it moves closer (Fig. 5A). Likewise, C2.36 presents a heterocyclic nitrogen that can potentially form a H-bond with E267, similarly, provided a protonation or water bridge is present (Fig. 5B). In addition, C2.36 forms a T-shaped π-stacking interaction with H574, whose sidechain has rotated 90 degrees (compared to DrOTOP1_apo), presumably to accommodate the interaction (Fig. 5B). Considering other inhibitors discovered from the second round of screening, many contain functional groups like the carbonyl groups of Group 1 molecules and the heterocyclic nitrogen atoms of Group 2 molecules (Supplementary Fig. 3E), that can potentially form similar H-bonds with E267 as C2.2 and C2.36, assuming the binding modes are similar. In the central site, C2.2 binds with the following interactions: hydrophobic packing with residues V201, F271, A274, V278, and A388, π-stacking with F277, and H-bonding with E384 (Fig. 5C).

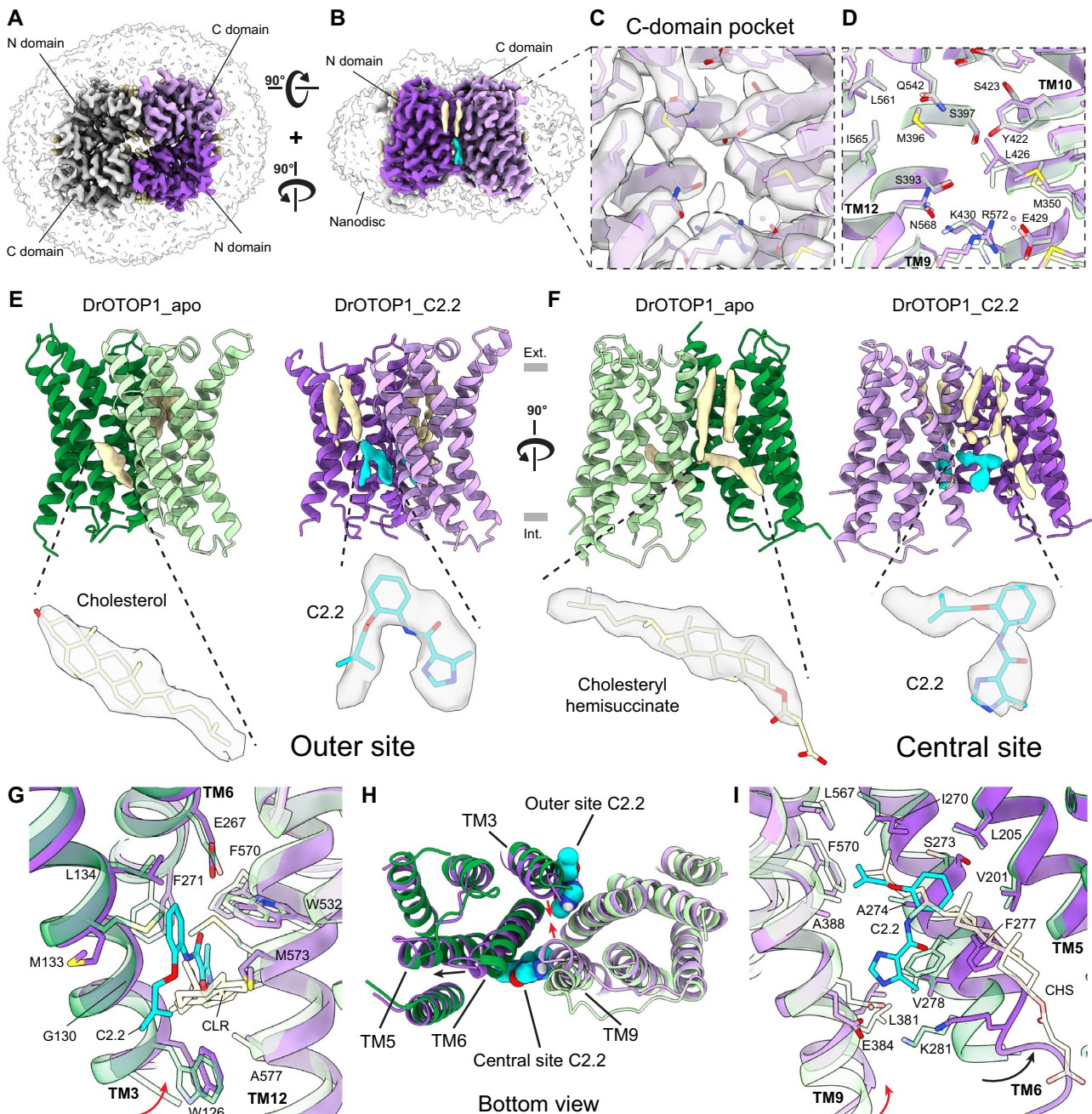

**Fig. 3 | Cryo-EM structure of DrOTOP1 with inhibitor C2.2. A** Top and **B** side views of cryo-EM structure of DrOTOP1 complexed with C2.2 (DrOTOP1_C2.2), with one monomer colored purple (N-domain) and plum-pink (C-domain), the other monomer colored light gray, bound-lipids colored tan, small molecules colored cyan, and the nanodisc micelle shown as a transparent surface. **C** Inset of (**B**) zoomed on the C-domain pocket with the model colored in plum-pink and cryo-EM map shown as a gray transparent surface. **D** Same view and coloring as (**C**) showing DrOTOP1_C2.2 in comparison with the apo structure of DrOTOP1 (DrOTOP1_apo) (PDB:6NF4) super-imposed and shown as a transparent light green stick and cartoon, with pocket-lining residues labeled. **E** Side views of DrOTOP1_apo (left) and DrOTOP1_C2.2 (right) models with only the cryo-EM density of cholesterol lipids (tan) and inhibitor C2.2 (cyan) near the outer site shown. Insets show density found in the outer site as a transparent surface, cholesterol (tan) and with C2.2 (cyan) models shown as sticks. Gray rectangles

indicate the height of the lipid bilayer, with the extracellular side (Ext.) on top and the intracellular side (Int.) below. **F** 90° rotated view of models as (**E**), showing cryo-EM densities of cholesterol lipids (tan) and C2.2 (cyan) near the central site. Insets show density found in the central site as a transparent surface, with cholesteryl hemi-succinate (CHS) (tan) and C2.2 (cyan) shown as sticks. **G** View of outer sites showing a superimposed comparison of DrOTOP1_C2.2 with bound C2.2 and DrOTOP1_apo with bound cholesterol (CLR), with DrOTOP1_apo shown as transparent stick and cartoon, highlighting conformational differences. Movement of TM3 is indicated (red arrow). **H** Bottom view of DrOTOP1_C2.2 and DrOTOP1 superimposed, with C2.2 models shown as cyan spheres. Movement of TM3 and TM9 indicated with red arrows, and movement of TM6 indicated with a black arrow. **I** Similar view as (**G**) for the central site, with C2.2 and CHS shown in the central site, highlighting conformational changes. Movement of TM6 (black arrow) and TM9 (red arrow) are indicated.

Each non-aromatic residue was mutated to tryptophan, which was expected to cause steric occlusion. We chose this approach as it was successful in identifying a mutation that affected the activity of inhibitors in our earlier experiments (Supplementary Fig. 5B, C, E, F), and because it

does not rely on the accuracy of the binding pose or specific protein-ligand interactions. This seemed appropriate given the small number of H-bonds in these pockets, the ambiguity of how the inhibitors fit the density, and the prevalence of hydrophobic packing interactions (Fig. 5A–C).

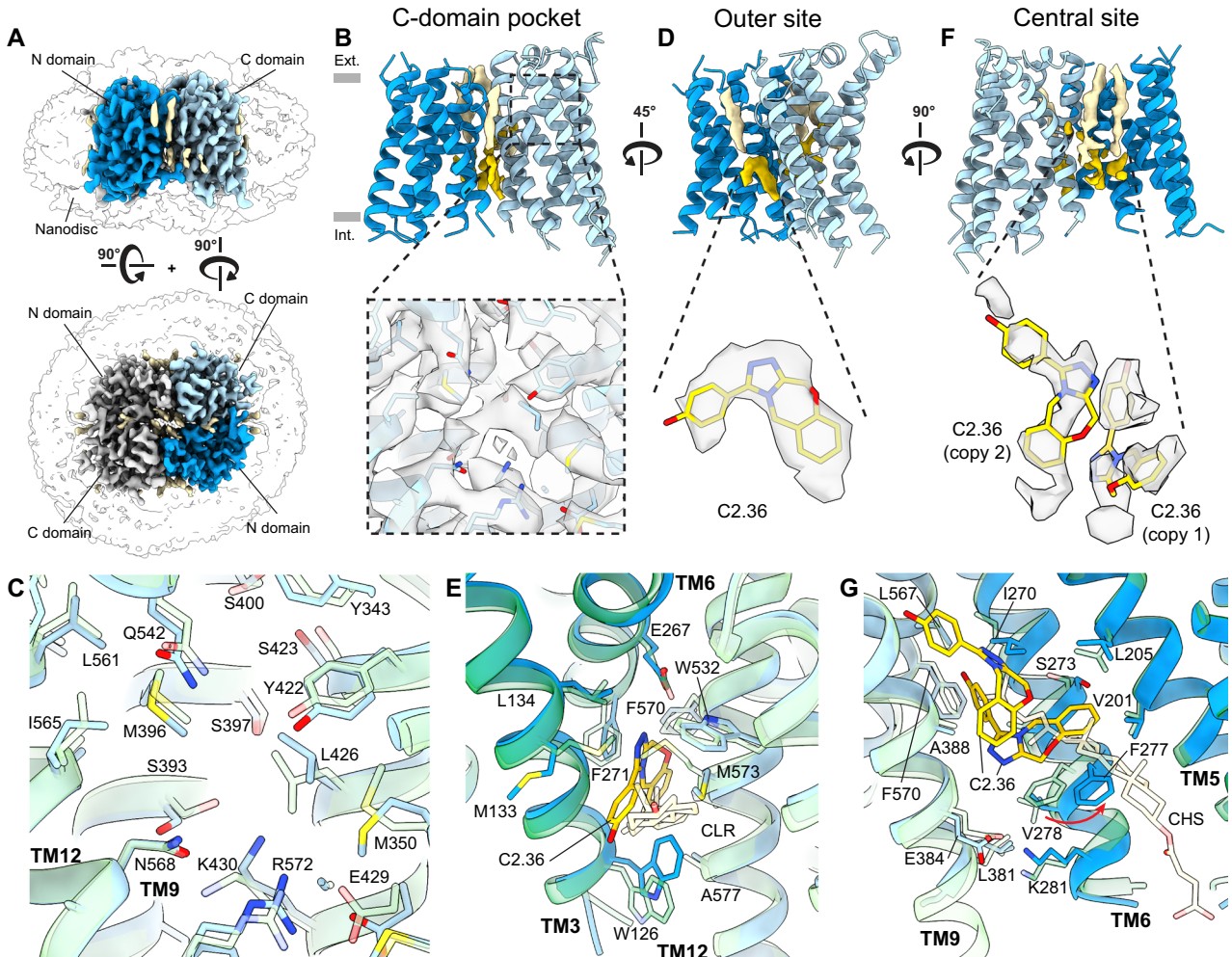

**Fig. 4 | Cryo-EM structure of DrOTOP1 with inhibitor C2.36. A** Side (top) and top (bottom) views of cryo-EM structures of DrOTOP1 in complex with C2.36 (DrO-TOP1_C2.36), with one monomer colored blue (N-domain) and light blue (C-domain), the other monomer colored light gray, bound-lipids colored tan, small molecules colored gold, and the nanodisc micelle shown as a transparent surface. **B** Side view of the DrOTOP1_C2.36 model, with density for bound lipids and small molecules shown. The inset shows the C-domain pocket with the cryo-EM map shown as a gray transparent surface. Gray rectangles indicate the height of the lipid bilayer, with the extracellular side (Ext.) on top and the intracellular side (Int.) below. **C** View of the C-domain pocket with DrOTOP1_apo (transparent) and DrOTOP1_C2.36 aligned and superimposed, with pocket lining residues labeled and shown as sticks, and backbones shown as cartoons to highlight conformational differences or the lack thereof. **D** 45° rotated side view from (**B**) shows the outer site, with the inset showing non-protein density found in the outer site as transparent surfaces, with the inhibitor C2.36 modeled in, shown as sticks. **E** View of outer site comparing DrOTOP1_C2.36 with C2.36 bound and DrOTOP1_apo with cholesterol (CLR) bound. Pocket lining residues are labeled and shown as sticks. **F** 90° rotated view from (**D**) shows the central site, with the inset showing non-protein density found in the central sites as transparent surfaces, with inhibitors C2.36 modeled in the density, shown as sticks. **G** View of central site comparing DrOTOP1_C2.36 with two copies of C2.36 bound and DrOTOP1_apo with CHS bound. Pocket lining residues are labeled and shown as sticks. Movement of the side chain of F277 is indicated with a red arrow.

First, we tested mutations of the outer site (G130W, M133W, and M573W) near the IC$_{50}$ of C2.2. Among the outer site mutations, G130W and M573W significantly reduced inhibition by 10 μM C2.2 (Fig. 5D). Dose-response data revealed right-shifted IC$_{50}$ values of 130.4 μM for G130W and 30.7 μM for M573W (Fig. 5E, F). Both mutations decreased the Hill coefficient to ~1.2 (compared with 1.5 in WT). We also tested L134W, another mutation in the outer site, which showed dramatically slowed activation kinetics to acid stimuli, suggesting that the mutation affects channel gating (see below). L134W appeared to be completely insensitive to 10 μM C2.2 (Fig. 5G, H) and exhibited the same current kinetics as the vehicle control (note that in both cases, our method of measuring current inhibition by examining off responses appears to show a ~20 inhibition of the currents).

We next tested mutations of the central site (L205W, A274W, F277A, E384A L385W, A388W). V201W and S389W, two additional central site mutations, were excluded from the study due to complete

loss of channel activity. Mutation of E384 to alanine is expected to disrupt potential hydrogen bonding with C2.2. Of note, E384 is one of the few charged residues in the central pocket and could form a salt bridge with K281, that could also be affected by the mutation. Additionally, in our DrOTOP1_C2.2 structure, we observed that F277 underwent a substantial rearrangement, likely to avoid steric clash and accommodate C2.2 binding. Thus, we introduced the F277A mutation to test if removing this steric clash would enhance C2.2 inhibition. Among these mutations, only E384A altered inhibition by 10 μM C2.2, reducing block to ~40% compared to ~67% in wildtype DrOTOP1, and shifting the IC$_{50}$ to 15.3 μM and the Hill coefficient to ~1.2 (Fig. 5D, F). That E384A remained functional indicates that the proposed salt bridge is not essential for channel function, although changes in gating were observed (see below).

Given our limited success in identifying mutations of the central site that disrupt antagonism by C2.2, we tested whether mutations of

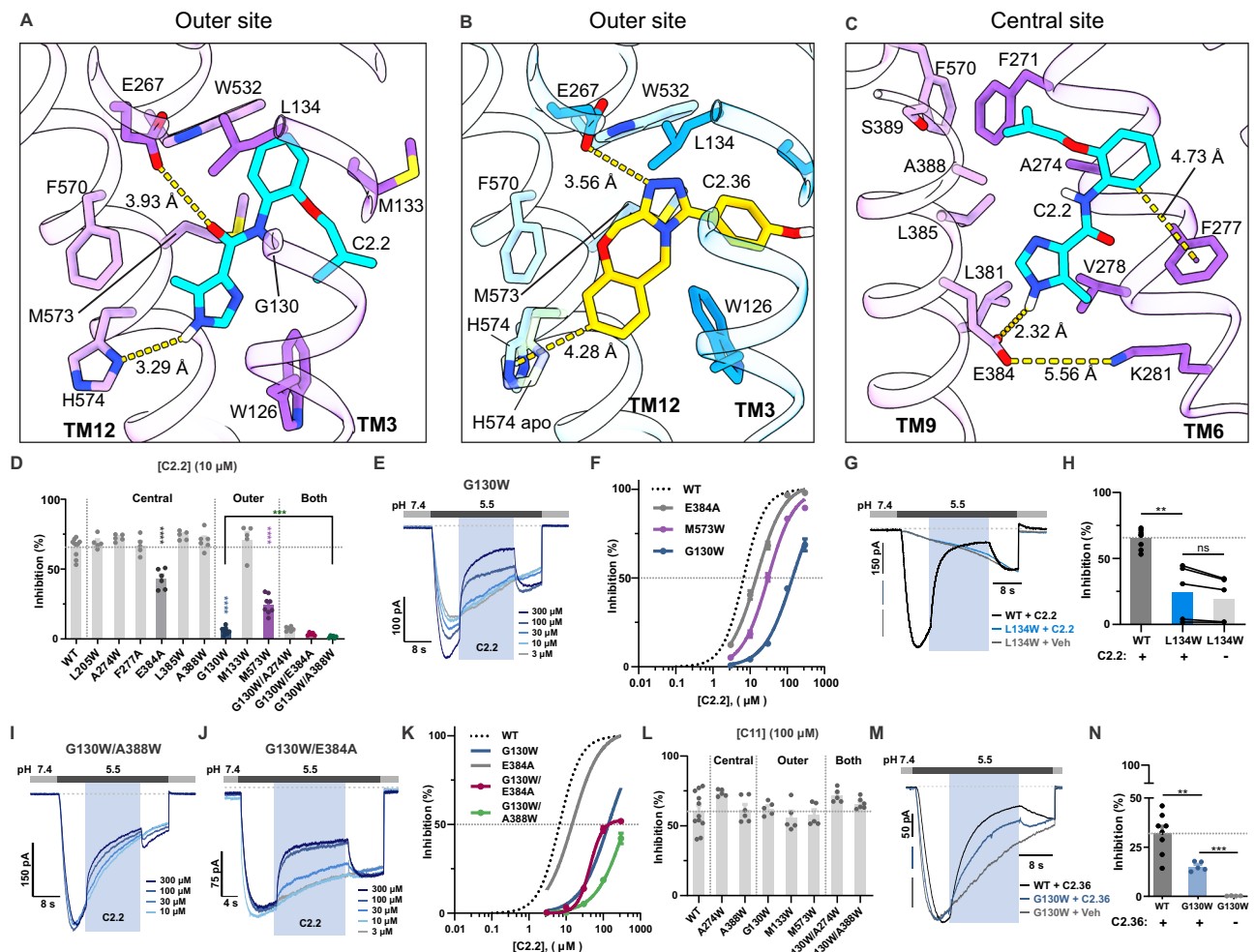

**Fig. 5 | Effects of mutagenesis of the outer and central sites on activity of inhibitors. A** View of C2.2 (cyan) bound in the outer site. Distances between atoms are shown as yellow dashed lines. **B** Same view as (**A**) for C2.36 (gold) bound in the outer site of DrOTOP1_C2.36. H574 from DrOTOP1_apo is shown as a transparent stick. **C** View of C2.2 (cyan) bound in the central site of DrOTOP1_C2.2. **D** Inhibition by 10 μM C2.2 in cells expressing mutations as indicated ($n = 5$–9 cells per construct). **E** Representative traces showing dose-dependent inhibition of G130W by C2.2. **F** Dose-response data for inhibition by C2.2 for mutants G130W, E384A, and M573W, fitted to the Hill equation ($n = 8$, 7, and 5 cells, respectively); WT data is from Fig. 2F. IC$_{50}$s and Hill coefficients: E384A: 15.2 μM, $h = 1.18$; M573W: 30.7 μM, $h = 1.24$; G130W: 130.4 μM, $h = 1.15$. **G** Representative traces showing the lack of inhibition in L134W (blue) by 10 μM C2.2 and vehicle control (0.01% DMSO, gray), compared with the effect of WT (black). **H** Inhibition by C2.2 or vehicle from experiments as in G measured from recovery of current (WT; $n = 10$ cells, L134W; $n = 5$ cells). **I, J** Representative traces showing dose-dependent inhibition by C2.2

in G130W/A388W and G130W/E384A mutants. **K** Dose-response data for inhibition by C2.2 for mutants G130W/A388W and G130W/E384A, fit to the Hill equation ($n = 6$ and 8 cells); G130W, E384A, and WT data from (**F**) and single mutant data from Fig. 2F. IC$_{50}$s and Hill slopes: G130W/A388W: 397.4 μM, $h = 1.10$; G130W/E384A saturates at ~ 50%. **L** Inhibition by 100 μM C11 in cells expressing central, outer, or double mutants ($n = 5$–11 cells). **M** Representative traces showing inhibition by 100 μM C2.36 of WT (black) and G130W (blue), compared with the effect of vehicle control (0.1% DMSO). in gray. **N** Inhibition by C2.36 (WT and G130W) or vehicle (G130W) (WT; $n = 8$ cells, G130W + C2.36; $n = 5$ cells, G130W + Veh; $n = 4$ cells). Significance determined by one-way ANOVAs with Dunnett's correction (mutants vs. WT for **D, L**; double mutants vs. G130W for **D**), unpaired two-tailed $t$ tests with Welch correction (**H, N**) and paired two-tailed $t$ test (**H**; C2.2 vs. vehicle on L134W). *$P < 0.05$, **$P < 0.01$, ***$P < 0.001$, ****$P < 0.0001$ for all statistical tests. All data presented as mean ± SEM. Experiments performed at $V_m = -80$ mV. Source data are provided as a Source Data file.

the central site in a background of G130W (the most impactful outer site mutation) might unmask contributions of this site. Thus, we generated and tested double mutants of G130W with E384A, A274W, and A388W. Among these double mutants, only G130W/A388W showed a statistically significant difference in inhibition by 10 μM C2.2 compared to G130W alone (Fig. 5D). We also measured the dose-dependence of inhibition by C2.2 for both G130W/A388W and G130W/E384A (Fig. 5I, J). The G130W/A388W mutation shifted the IC$_{50}$ to 397 μM, a ~ 3-fold reduction in sensitivity as compared with the single mutant (Fig. 5I, K). The G130W/E384A mutant showed a similar IC$_{50}$ to G130W, but inhibition saturated at ~ 50% by 100 μM C2.2 with no further inhibition observed at 300 μM (Fig. 5J, K). G130W was tested in parallel to confirm that 100 and 300 μM C2.2 produced dose-dependent inhibition. In contrast, none of the

mutants, including G130W, showed any change in sensitivity to C11, tested at 100 μM (Fig. 5L).

To confirm our structural data showing that C2.36 binds to the outer site, we tested the G130W mutant. WT DrOTOP1 currents evoked in response to a pH 5.5 stimulus are slowly inhibited by C2.36, which is poorly reversible (Fig. 5M). Compared to WT, the G130W mutant showed significantly less inhibition and faster wash-off of C2.36 (Fig. 5M, N), consistent with the interpretation that the G130W mutation disrupts binding of C2.36 in the outer site. Other mutants were not tested, given the difficulty in assessing efficacy for this poorly reversible inhibitor.

Together, these results provide functional evidence that mutations in both the outer and central sites reduce inhibition by C2.2, confirming that these conserved cholesterol binding sites double as the binding sites for inhibition by C2.2 of DrOTOP1.

**Mutations in the central and outer sites alter gating of DrOTOP1**

While testing mutations in all three pockets, we observed that some mutations that changed sensitivity to C2.2 also affected activation kinetics of the currents in response to acid stimuli, suggesting potential changes in channel gating. (Supplementary Fig. 5H and Fig. 5G). To further explore the effects of the mutations on gating, we measured responses of the channels over a broad pH range, including both acidic and alkaline stimuli (pH 10-pH 5.0). As a basis for comparison, we first measured responses from wildtype DrOTOP1. As shown in Fig. 6A, WT DrOTOP1 is steeply activated by acid stimuli beginning ~ pH 6.0 and is relatively insensitive to the alkaline stimulus (pH 10; but note the large tail current).

Examination of the response properties of five mutants with decreased sensitivity to C2.2 revealed that they fell into three groups: (1) mutants that showed increased sensitivity to acidic stimuli (E384A, in the central site and I565W, in the C-domain pocket) (2) mutants that showed reduced sensitivity to acidic stimuli (L134W and M573W, both in the outer site) and (3) one mutant that showed no change in pH-dependent gating (G130W, in the outer site) (Fig. 6B–G). Changes in sensitivity to the pH stimulus were reflected in both the magnitude of the currents as normalized to the response to the pH 5.0 stimulus (Fig. 6G) and their kinetics. For example, in response to a pH 5.5 stimulus, I565W currents reached a peak faster than WT DrOTOP1 (Supplementary Fig. 5H and Fig. 6E), while L134W currents were much slower, and did not reach a peak during the 8 s stimulus (Figs. 5G and 6C). Moreover, mutants that were less sensitive to acid stimuli (L134W) showed reduced responses to the alkaline stimulus, while mutants (E384A and I565W) that were more sensitive to acid stimuli showed larger responses to the alkaline stimulus. Fitting data from normalized responses over the entire pH range showed that the $pH_{50}$ (pH at which currents were at ½ their projected maximal value) showed similar shifts in pH sensitivity: WT: $pH_{50}$ = 5.64, G130W: pH 5.68, L134W: pH 5.43, E384A: pH 5.97, I565W: pH 6.36, and M573W: pH 5.43 (Fig. 6G, inset). In addition, we noted that the kinetics of a subset of these channels showed two phases of activation, which is an indication of two distinct gating mechanisms (e.g., opening of two pores), which may be independently affected by the mutations (Fig. 6D–F). Despite these changes in channel gating, inhibition by 1 mM $Zn^{2+}$ was similar for all mutants tested (Fig. 6H).

## Discussion

From a small catalog of ~ 300,000 molecules[44], we discovered several inhibitors for DrOTOP1 by virtually screening against a pocket with no known ligands. By doing so, we circumvented the lack of established high-throughput functional assays for Otopetrins and identified functional molecules using manual whole-cell patch-clamp, culminating in the identification of C2.2, an inhibitor with an $IC_{50}$ of 6.67 μM. Our second round of screening exhibited clear enrichment in inhibitory activity (Fig. 2A, D), indicating that our analog-by-catalog search successfully identified a chemical pattern of inhibitors against DrOTOP1. Our cryo-EM structures revealed a binding site (outer site) on the intrasubunit interface, which we validated by showing that site-directed mutagenesis of residues therein disrupted the inhibitory activity of C2.2 by as much as 20-fold. A second site (central site) was also identified on the structure, which was less susceptible to perturbation through site-directed mutagenesis. We also identified a slower-acting inhibitor, C2.36, which, based on limited structural and functional data, appears to bind to the same sites.

Inspection of molecules from the second screen that achieved more than 25% inhibition at 200 μM reveals that they adopt a loosely planar "L" shape (Supplementary Fig. 3E). While it is possible that the ChemBridge library[44] has a prevalence of molecules that match this shape, we hypothesize that the emergence of this shape indicates a basic structure-activity relationship pattern, assuming that they function through the same binding sites as C2.2 and C2.36

(Figs. 3E, F and 4D, F). Here, we note that our Group 2 inhibitors (Supplementary Fig. 3E) were filtered based on structural similarities to FDA-approved drugs containing dibenzazepine cores, like oxcarbazepine, which belong to the larger family of tricyclic antidepressants (TCAs). TCAs are known to broadly reduce taste reception[49] and non-specifically inhibit various ion channels, like sodium[50] and calcium channels[51,52]. Even so, given the chemical structure similarities seen here, it may be possible that some TCAs or other tricyclic compounds may inhibit DrOTOP1 or other Otopetrins through the intrasubunit sites. Further testing of TCAs and similar compounds could investigate this hypothesis.

Our cryo-EM structures of DrOTOP1_C2.2 and DrOTOP1_C2.36 serendipitously revealed the inhibitors bound to unexpected sites. In hindsight, finding the inhibitors to bind in different sites than the initial docking location is not surprising[53,54] and highlights the importance of diligent experimental validation of virtual screening results[55,56] and thorough investigation of cryo-EM maps. Instead of binding to the C-domain pocket, the inhibitors bound in two distinct sites in the intrasubunit interface (Figs. 3E, F and 4D, F). For C2.2, binding at both sites was clearly resolved and supported by functional data (Fig. 5D, F, K), while for C2.36, functional data support binding in one site (Fig. 5M, N). The intrasubunit sites are both predominantly hydrophobic, hence, it seemed counterintuitive that ligands selected for a hydrophilic C-domain pocket instead bound to the intrasubunit sites. We suspect this is due to the ChemBridge library containing drug-like molecules that tend to be more lipophilic. In addition, the C-domain pocket is smaller and favors ligands with lower molecular weights, which can bind to the larger intrasubunit sites with minimal restrictions. Docking C2.2 and C2.36 into the intrasubunit sites produced poses resembling the cryo-EM poses, albeit with worse scores than the best scoring poses (Supplementary Fig. 9). We hypothesize this shortcoming is due to how the current scoring functions of AutoDock-GPU[48] or AutoDock Vina[43,57] do not consider significant contributions from the lipid bilayer[58] when docking into protein-lipid interface regions of membrane proteins, like the intrasubunit sites. At these sites, the ligands bind and replace cholesterol, which is similarly lipophilic and planar (Fig. 3E, F and Supplementary Fig. 3E). Our results therefore support a potential role for endogenous cholesterol to regulate the channels, as was previously hypothesized[8]. Both intrasubunit sites also harbor a few titratable residues, E267 and H574 in the outer site and E384 and K281 in the central site, which could provide key H-bonds with inhibitors (Fig. 5A–C). The protonation and deprotonation of these residues or the inhibitor molecules, when exposed to acidic conditions during channel activity, could influence the strength of their interactions. The slight reduction in inhibition by C2.2 observed when the pH is reduced to 5.0 (Fig. 2I) could allude to such an effect of side-chain titration. However, we do not expect C2.2 itself to have titratable groups within the pH range considered in this study.

Through extensive structure-guided mutagenesis, we show that mutations in both the outer and central sites are capable of attenuating inhibition by C2.2. Mutations of G130, L134, and M573, located within the outer site, and E384 in the central site, all attenuated inhibition by C2.2. A double mutation of G130 and A388 shifts the $IC_{50}$ relative to G130 alone, suggesting that the contribution of A388 is unmasked only when binding at the outer site is impaired. Surprisingly, a double mutation of G130 and E384 reduced the maximal efficacy of C2.2 to ~ 50% at saturating concentrations, without further shifting the $IC_{50}$ relative to G130 alone. This result could be explained if the E384A mutation changed C2.2 into a partial antagonist, but this is not what we observed in the single mutant. Thus, our findings suggest a complex mechanism for inhibition by C2.2 that depends on interactions at both the outer and central sites and may involve allosteric changes in channel gating. In contrast to C2.2, we observed little indication of C11 binding in either the C-domain pocket or the intrasubunit sites by either site-directed mutagenesis or cryo-EM. It is possible that C11

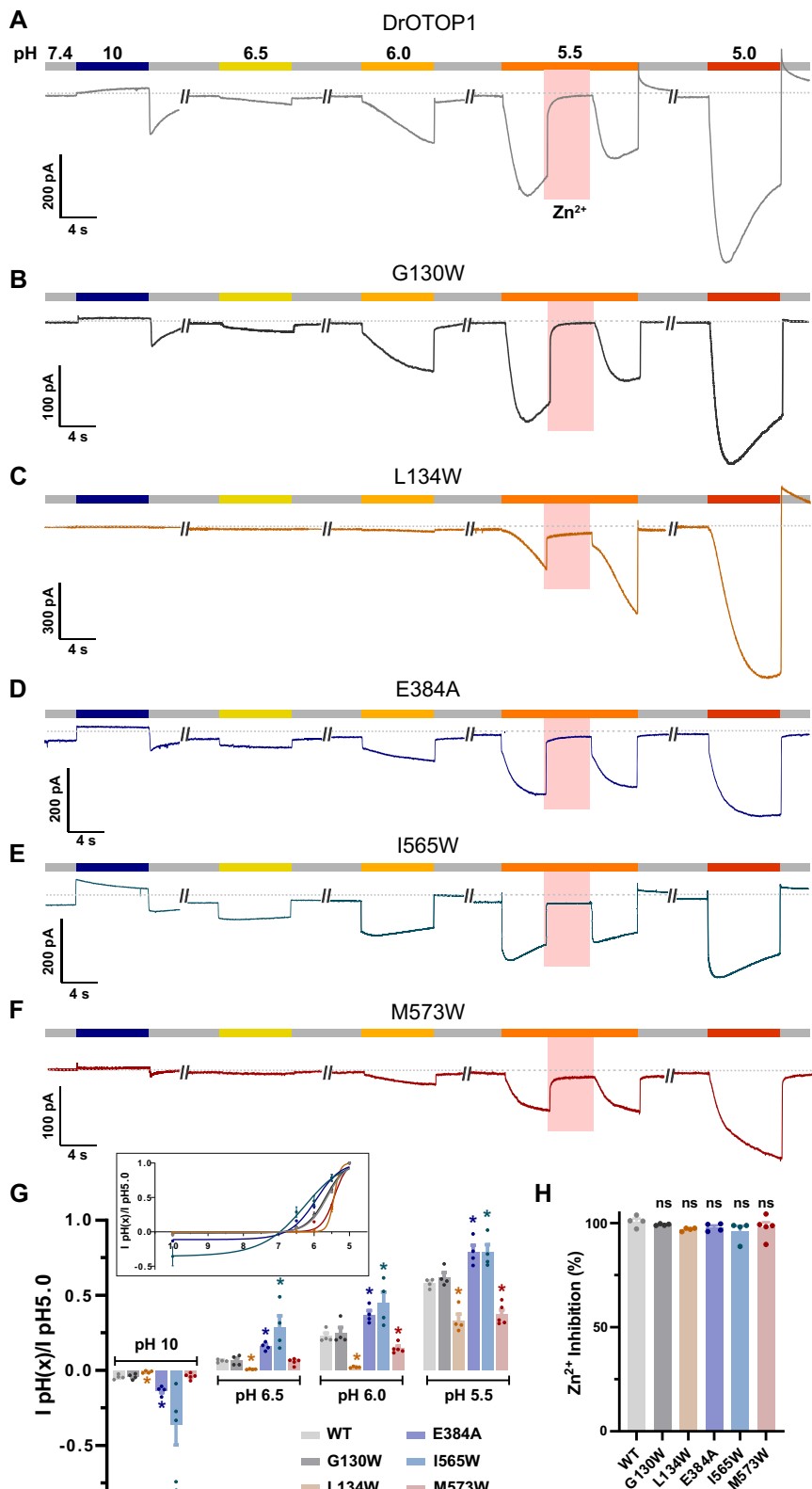

**Fig. 6 | Outer, central, and C-domain pocket mutants that reduce inhibitor efficacy also alter channel gating. A–F** Representative traces of currents from HEK-293 cells expressing WT (**A**), G130W (**B**), L134W (**C**), E384A (**D**), I565W (**E**), and M573W (**F**) over a pH range of 10 to 5.0. 1 mM $Zn^{2+}$ (pH 5.5) was applied during the pH 5.5 stimulus as indicated. **G** Current magnitudes normalized to response to pH 5.0, from experiments as in (**A–F**) ($n = 4$-5 cells per construct). Inset: Fitted $pH_{50}$ curves for WT and mutant channels. $pH_{50}$s were as follows. WT: pH 5.64, G130W: pH 5.68, L134W: pH 5.43, E384A: pH 5.97, I565W: pH 6.36, M573W: pH 5.43.

**H** Inhibition by 1 mM $Zn^{2+}$ from experiments as in (**A–F**), ($n = 4$-5 cells per construct). Significance was determined in (**G**) using non-parametric two-tailed Mann-Whitney tests to compare cumulative ranks to WT DrOTOP1 at the same pH. Significance was determined in (**H**) using one-way ANOVA with Dunnett's correction compared to WT DrOTOP1. All data presented as mean ± SEM. *$P < 0.05$, **$P < 0.01$, ***$P < 0.001$, ****$P < 0.0001$ for all statistical tests. Experiments were performed by holding cells at $V_m = -80$ mV. Source data are provided as a Source Data file.

**Table 1 | SMARTS patterns**

| Nickname | ID | SMARTS strings |
|---|---|---|
| 6-6 ring | 2 | [rD2]1:[r]:[r]:[r](:[r]:[r]:1)-[*]-[#6](=[#8])~[r]1:[r]:[r]:[r]:[rD2]:[r]:[r]:1 |
| 5-6 ring | 3 | [r]1:[r]:[r]:[r](:[r]:1)-[*]-[#6](=[#8])~[r]1:[r]:[r]:[r]:[rD2]:[r]:[r]:1 |
| 6-5 ring | 4 | [rD2]1:[r]:[r]:[r](:[r]:[r]:1)-[*]-[#6](=[#8])~[r]1:[r]:[r]:[r]:[r]:[r]:1 |
| 5-5 ring | 5 | [r]1:[r]:[r]:[r](:[r]:1)-[*]-[#6](=[#8])~[r]1:[r]:[r]:[r]:[r]:[r]:1 |
| Dibenzazepine core | 7 | [r]1-[r](:[r](-[r]-[r]-[r](:[r]-1:[r]:[r]):[r]):[r]):[r]:[r] |

binds one or more of these sites, but its binding could not be captured by cryo-EM or perturbed by single residue mutations, perhaps due to lower affinity or smaller size. Alternatively, C11 may bind and act through alternative site(s) than the ones we considered in this study.

Previously, extracellular or intracellular loops in OTOP channels have been implicated in channel gating[16,22–25]. More recently, a deep site within mOTOP1, S134 (the equivalent of G130 in DrOTOP1) (Supplementary Fig. 4), was shown to be necessary for activation of the channels by carvacrol[59]. Thus, the same site in mouse and in zebrafish OTOP1 appears to play a role in mediating responses to small molecules that in one case (mouse) activate the channels and in the other (zebrafish) inhibit activity. Furthermore, mutation of L134, E384, and M573 to tryptophan all resulted in changes in the pH sensitivity and activation kinetics of the currents. These results support the idea that the outer and central sites are part of the gating apparatus and that these deeper regions within the channel (intrasubunit interface) can influence gating in response to small molecules.

For OTOP channels, it remains unclear whether the pore is distinct from the gating apparatus or if these functions are one and the same. It is possible that changes in gating could affect the efficacy of the compounds[60], if they act as gating modifiers rather than pore blockers, a possibility we cannot rule out, one that is difficult to assess for these proton-activated proton channels[22]. This may be the case for the I565W mutation, which almost certainly acts at a distance to modify the efficacy of C2.2. Thus, while the structural basis for gating and permeation of OTOP1 channels remains obscure, taken together, our structural and functional data, including the observation that mutations in both the outer and central sites alter channel gating, favor the interpretation that C2.2 acts through an allosteric mechanism rather than by directly occluding the channel pore.

By identifying inhibitors for DrOTOP1, our work serves as proof-of-concept for virtual small molecule discovery of channel modulators for Otopetrins. Inhibitor C2.2 will be useful for structure-function studies on DrOTOP1, the sole structural representative ortholog for OTOP1, and could be developed into a more effective tool for use in physiological studies of OTOP1 in zebrafish. While the inhibitors are specific to zebrafish OTOP1, similar virtual screening efforts could yield channel modulators for mammalian orthologs of Otopetrins. Such studies could utilize the growing list of high-resolution cryo-EM structures of Otopetrins or predictive models generated by AlphaFold[61], which can accurately model the Otopetrin fold[22]. As the intrasubunit binding sites harbor highly conserved residues across Otopetrins (Supplementary Figs. 4 and 7B), pharmacologically targeting the same sites in other Otopetrins seem possible. Furthermore, our results show that C11 weakly blocks hOTOP1 currents at 300 μM (Fig. 2G), hence C11 could provide a basic framework for the discovery of inhibitors for mammalian OTOP1. Beyond inhibition, as the mechanisms of gating and proton sensing are further investigated and key extracellular loops[22], residues like H229 and D570 (in hOTOP1)[24], R292 in mOTOP1[16], and H234, E238, H531, and E535 (in mOTOP3)[25] are identified, more regions of Otopetrins will become viable to target for activation and gating modulation with small molecules. Potent and specific modulators for mammalian Otopetrins will bolster the arsenal of molecular tools with which to study these less-understood proton

channels in the context of physiology and pathophysiology in the future.

## Methods

### Receptor preparation for virtual screening
A monomer of the PDB model of DrOTOP1 (PDB: 6NF4, chain A)[8] was extracted and prepared for docking following the standard AutoDock protocol[62], by stripping all non-standard amino acids and bound ligands. Explicit hydrogens were added using Reduce[63], and the model was converted to PDBQT format using the "prepare_receptor.py" script from MGLTools v1.5.7[64]. The prepared receptor model of DrOTOP1 was then used for all docking-related analysis, unless otherwise specified.

For the second round of virtual screening, the side chain of F342 of DrOTOP1 was changed to a rotamer which increased the pocket volume, to allow for favorable docking of larger ligand molecules. The F342 rotamer-mutated model was prepared for docking following the same steps as described above.

### Binding site analysis
AutoSite v1.0[42] was used with default settings to identify potential small molecule binding sites on the prepared receptor model of DrOTOP1. Through visual inspection of the results, putative pockets in three locations – N-domain putative pore, intrasubunit interface, and C-domain putative pore – were considered suitable for virtual screening. The C-domain pocket was ultimately chosen for virtual screening based on shape, size, buriedness, a balance of hydrophobic and hydrophilic profile, and the presence of a functionally important salt bridge E429-R572.

### Chemical library preparation
For all virtual screening rounds, a 90% diversity set of the ChemBridge library[44] consisting of 302,893 drug-like molecules was used, the preparation for which was performed according to the standard AutoDock protocol[62] as described previously[65].

For the second round of virtual screening, SMARTS were designed and visualized using the SMARTS PLUS web-service[66], based on the chemical structure of C11 and the dibenzazepine family of FDA-approved drugs. The SMARTS strings are listed in Table 1. The SMARTS were used as queries for a substructure-based search of the full ChemBridge library[44] of ~1.3 M drug-like molecules (accessed 2021), which include the CORE Library Stock and the EXPRESS-Pick stock, using RDKit v2021.09.3[46], resulting in a filtered set of 35,908 molecules that matched the query. For this filtered set, tautomers were generated, and 3D coordinates were assigned using RDKit v2021.09.3[46] and molscrub v0.1.1[67]. The molecules were then converted to PDBQT format using Meeko v0.6.1[68].

Representative "cloud" image of the ChemBridge library molecules shown in Supplementary Figs. 1 and 3D was generated with molcloud v0.3.0[69], using the SMILES of all molecules purchased for testing.

### Molecular docking and analysis
For the first round of virtual screening, following the standard AutoDock protocol, an AutoDock Vina[43] configuration file was written to

describe a docking box centered at coordinates $x = 119.433$, $y = 117.022$, $z = 110.976$, and of size $23.25 \times 23.25 \times 23.25$ Å, using the graphics user interface (GUI) of AutoDock Tools (ADT), available as part of MGLTools v1.5.7[64], which fully encapsulated the putative pocket identified by AutoSite v1.0[42] in the C-domain of DrOTOP1, including the side chains of pocket-lining residues. Using the configuration file, molecular docking calculations were performed with AutoDock Vina v1.1.2[43], with default settings (exhaustiveness = 8), on 302,893 molecules from the ChemBridge Library[44] (90% diversity set). Docking results were filtered using Python scripts from AutoDock Raccoon2 v1.5.7[62,70] to select for molecules with predicted scores better (lower value) than −7.5 kcal/mol, and with at least one predicted H-bond. The resulting 35,779 molecules were clustered based on maximum common substructures (MCS) and molecular fingerprint based Tanimoto similarity[71] using an RDKit v2021.09.3[46] implementation of the MultiMCS clustering algorithm[72]. In parallel, the molecules were also clustered using the dimensionality-reduction method UMAP v0.5.9[73] and HDBSCAN v0.8.39[74] clustering, based on their molecular fingerprints. The molecules were selected for visual inspection in PyMOL v2.5[75], top-down in order of their predicted score, while selecting no more than three molecules from each cluster of any clustering method, in order to enrich for chemical diversity. During visual inspection, small molecules were evaluated based on the validity of their docked poses, favorable protein-ligand interactions, proper structural conformation, potential desolvation cost, desirable chemical functional groups, uniqueness of chemical structure, and availability of compound stock from ChemBridge[44]. A final set of 50 molecules were selected and purchased.

The second round of virtual screening was performed similarly as described above, using the F342 rotamer-mutated model of DrOTOP1 as the receptor, the SMARTS filtered set of 35,908 molecules as the chemical library, and AutoDock Vina v1.1.2[43] as the docking calculation engine, using the same configuration file and docking box. Using the same analysis steps as described above, a final set of 51 molecules were selected and purchased. The ChemBridge IDs and SMILES for all compounds selected and purchased are listed in Supplementary Table 1.

Docking C2.2 and C2.36 into the intrasubunit outer sites was performed using AutoDock-GPU v1.6[48]. Chain "A" residues of DrOTOP1_C2.2 and DrOTOP1_C2.36 (aligned to DrOTOP1_C2.2) structures were trimmed of ligands and prepared as receptors for docking using Meeko v0.6.1[68]. C2.2 and C2.36 were parametrized and converted to PDBQT files as described above. AutoDock grid parameter files (gpf) encoding a docking box centered at coordinates $x = 130.070$, $y = 108.131$, $z = 102.373$, and of size $48 \times 58 \times 68$ npts (0.375 Å spacing) for the outer site, and a docking box centered at $x = 117.070$, $y = 108.131$, $z = 102.373$ and of size $52 \times 58 \times 74$ npts (0.375 Å spacing) for the central site were generated using Meeko v0.6.1[68]. AutoGrid v4.2.6[76] was used to generate receptor grid maps in each docking box. Docking calculations were done using AutoDock-GPU v1.6[48], with default parameters, except the number of Lamarckian Genetic Algorithm (LGA) runs was set to 200 to produce more poses. The resulting docking log files (dlg) were converted to SDF files using Meeko v0.6.1[68], which were loaded into UCSF ChimeraX v1.10[77–79] for analysis of docked poses. When docking the second copy of C2.36 into the central site, the receptor was prepared from DrOTOP1_C2.36, but the first copy of C2.36 (cryo-EM modeled pose) was kept in the model and was treated as part of the receptor.

Chemical structures of small molecules and SMARTS were drawn in ChemDraw® v22.0.0[80] and SMARTS PLUS[66], respectively. Molecular docking figures were generated using PyMOL v2.5[75] and UCSF ChimeraX v1.10[77–79].

## Small molecule storage and preparation
Compounds were purchased from ChemBridge (San Diego, CA, USA) from hit2lead.com in quantities ranging from 10 mg – 5 μmol. In addition, three FDA-approved drugs, oxcarbazepine, eslicarbazepine acetate, and tianeptine – were separately purchased from common vendors. All small molecules used in the study are of high purity (> 85%) as guaranteed by ChemBridge[44] and otherwise meet the community requirements[81] to the best of our knowledge.

For electrophysiology, master stocks were made of each compound at 100 mM – 25 mM depending on individual solubility in 100% dimethyl sulfoxide (DMSO), given arbitrary numbers to be easily referenced, and aliquoted and stored at −20 °C until used for electrophysiology. When used for testing, compounds were thawed at room temperature and added at the appropriate concentrations for each experiment. When doing dose-dependent responses, serial dilutions were made from the highest tested concentration of each compound.

For cryo-EM sample preparation, all master stocks were prepared by dissolving in 100% DMSO at the following concentrations: C11 at 111 mM (24.98 mg/mL), C2.2 at 2.17 M (588.07 mg/mL) and C2.36 at 20 mM (5.58 mg/mL). All stocks were kept at −20 °C for storage.

## Cell culture, transfection for electrophysiology, and construct generation for whole-cell patch-clamp
HEK-293 cells (CRL-1573, ATCC) were cultured in DMEM (Thermofisher 11995073) containing 10% fetal bovine serum (Life Technology 16000044) and Penicillin-Streptomycin (100 I.U./mL) and (100 μg/mL), respectively. Cells were transfected with WT DrOTOP1 or mutant DrOTOP1 channels in 35 mm petri dishes, with ~ 500 ng DNA and 2 μL TransIT-LT1 transfection reagent (Mirus Bio Corporation, Cat # MIR2300) following the manufacturer's protocol. All constructs were subcloned into a pcDNA3.1 vector that adds an N-terminal fusion tag consisting of an octahistidine tag followed by eGFP, a Gly-Thr-Gly-Thr linker and a 3C protease cleavage site (LEVLFQGP) followed by WT DrOTOP1, WT mOTOP1, WT hOTOP1, or mutated DrOTOP1. Point mutations were generated via In-Fusion Cloning (Takara) using single-stranded DNA primers generated through IDT and sequence confirmed by Sanger sequencing (GeneWiz).

## Patch-clamp electrophysiology
Whole-cell patch-clamp recording was performed as previously described[6,22,25]. Recordings were made with an Axopatch 200B amplifier, digitized with a Digidata 1322a 16-bit data acquisition system, acquired with pClamp 8.2 and analyzed with Clampfit 10 (Molecular Devices). Recordings were sampled at 5 kHz and filtered at 1 kHz. Patch Pipettes with resistance of 2-5 mOhm were fabricated from borosilicate glass, and only recordings with a giga-Ohm seal were used in analysis. Membrane potential was held at −80 mV for the duration of recordings. At the beginning of recordings and between, cells were washed with Tyrode's solution (see below). Once a giga-Ohm seal was reached, cells were lifted and placed in front of microcapillary tubes controlled by a Fast-Step perfusion system (Warner Instruments) so that extracellular solutions could be rapidly exchanged during each experiment.

## Patch-clamp electrophysiology solutions
Standard Tyrode's solution consisted of 145 mM NaCl, 5 mM KCl, MgCl₂, 2 mM CaCl₂, 20 mM D-Glucose, and 10 mM HEPES (for pH 7.4). All other extracellular patch-clamp solutions contained 160 mM NMDG-Cl, 2 mM CaCl₂, and the following buffers: 10 mM HEPES (for pH 7.4), 10 mM MES (for pH 6.5-5.5), 10 mM CHES for pH 10, and 10 mM HomoPIPES for (pH 5.0). pH adjusted with HCl. When adding compounds to extracellular solutions, the concentration of DMSO in solution was never higher than 0.3%. The pipette contained intracellular solution made up of 120 mM Cs-aspartate, 15 mM CsCl, 2 mM Mg-ATP, 5 mM EGTA, 2.4 mM CaCl₂ (100 nM free Ca²⁺), and 10 mM HEPES with a pH of 7.3. All extracellular solution osmolarity was

adjusted to 300 mOsm and intracellular osmolarity was adjusted to 290 mOsm.

## Expression constructs and viral production

A previously generated codon-optimized full-length DrOTOP1[8] (Uni-Prot: Q7ZWK8) construct containing, in order, an 8x-His tag, eGFP, a Gly-Thr-Gly-Thr linker, and a 3C protease cleavage site (LEVLFQGP) at the N-terminal was cloned into a pEG BacMam plasmid[82] for BacMam expression. Bacmid DNA was generated from the DrOTOP1 pEG construct using DH10Bac™ cells following the Bac-To-Bac® Expression system protocol.

## Protein expression

The Bacmid DNA was then used to transfect Sf9 cells at $1 \times 10^6$ cells/mL density to generate P1 baculovirus in 6-well plates, which was incubated at 27 °C for 5 – 7 days. P2 baculovirus was generated by using P1 baculovirus to transduce Sf9 cells at $1.5 – 2.5 \times 10^6$ cells/mL density at a ratio of 1: 500 (v/v), which was incubated at 27 °C for 3 – 4 days in suspension in Erlenmeyer flasks. P2 baculovirus was harvested by pelleting the Sf9 cells by centrifugation at $4000 \times g$ and further centrifuging the resulting supernatant at $25,000 \times g$ to pellet the baculovirus. The baculovirus was then reconstituted into Human Embryo Kidney (HEK) 293F cell media supplemented with 2% Fetal Bovine Serum (FBS) and filtered using a 0.2 μm Stericup® filter and stored in the dark at 4 °C, before being used to transduce HEK293F cells at $1.5 – 2.5 \times 10^6$ cells/mL density at a ratio of 1: 10 (v/v) = baculovirus: HEK293F. Transduced HEK293F cells were grown under 8% CO2 for ~12 h at 37 °C, then supplemented with 10 mM sodium butyrate and grown for ~36–60 h at 32 °C. Cells were then pelleted by centrifugation at $4000 \times g$, resuspended in ice-cold 50 mL Phosphate Buffered Saline (PBS) and pelleted again by centrifugation at $4500 \times g$. The resulting cell pellets were used immediately for protein purification or stored at −80 °C for later use.

## Protein purification for cryo-EM sample

All protein purification steps were performed at 4 °C, and buffers were chilled on ice until use unless otherwise noted. Cell pellets were thawed in a room-temperature water bath and resuspended in Solubilization Buffer (25 mM Tris pH 8.0, 150 mM NaCl, 1% n-dodecyl-β-d-maltoside (DDM), 0.15% cholesteryl hemisuccinate (CHS), 2 μg/μL leupeptin, 2 μg/μL aprotinin, 1 mM phenylmethylsulfonyl Fluoride (PMSF), 2 μM pepstatin A and 2 mM dithiothreitol (DTT)) and was stirred for 1 – 3 h for detergent solubilization. The lysate was centrifuged at $40,000 \times g$ and the supernatant was collected. Anti-GFP nanobody[83]-coupled CNBr-Activated Sepharose 4B resin (Cytiva) (prepared in-house) was washed once with PBS and once with SEC1 Buffer (25 mM Tris pH 8.0, 150 mM NaCl, 0.07% DDM, 0.01% CHS, 0.4 μg/μL aprotinin, 0.4 μM pepstatin A and 2 mM DTT) and was added to the supernatant and the mixture was stirred for 2 – 3 h. The resin was then collected over a gravity chromatography column, washed once with 10 column-volume (CV) of Solubilization Buffer, once with 15 CV SEC1 Buffer and was resuspended in 3 CV SEC1 Buffer to make a ~25% slurry. 50 μg of PreScission protease (prepared in-house) was added to the resin slurry, and the mixture was rotated overnight. Next, the mixture was flowed over a gravity chromatography column, and flowthrough containing GFP-cleaved DrOTOP1 was collected and concentrated in a 100 kDa cutoff centrifugal concentrator (Amicon® Ultra or Pierce™) that was pre-equilibrated with SEC1 Buffer, to 500 μL volume. The sample was centrifuged at $21,100 \times g$ to remove insoluble debris before being injected into a Superose 6 Increase 10/300 column equilibrated with SEC1 buffer. The resulting peaks were analyzed with sodium dodecyl sulfate-polyacrylamide gel electrophoresis (SDS-PAGE), and fractions corresponding to dimeric DrOTOP1 were pooled. Detergent solubilized pure DrOTOP1 was mixed with membrane scaffold protein (MSP) 2N2 (made in-house or SIGMA) and soybean polar lipid extract (Avanti)

at a molar ratio of 15: 25: 200 = DrOTOP1: MSP2N2: lipids. Bio-Beads SM-2 Resin (Bio-Rad), washed once with 15 mL of MeOH, twice with 15 mL ddH$_2$O and pre-equilibrated with SEC2 Buffer (25 mM Tris pH 8.0, 150 mM NaCl, 0.4 μg/μL aprotinin, 0.4 μM pepstatin A and 2 mM DTT) was added to the sample mixture at a concentration of 200 mg of resin per mL of sample and was rotated overnight to remove detergents and form lipid nanodisc. The sample was separated from the biobeads and was concentrated in a 100 kDa cutoff centrifugal concentrator (Amicon® Ultra or Pierce™) that was pre-equilibrated with SEC2 Buffer, to a 500 μL volume. The sample was centrifuged at $21,100 \times g$ to remove insoluble debris before being injected into a Superose 6 Increase 10/300 column equilibrated with SEC2 buffer. The resulting peaks were analyzed with SDS-PAGE, and fractions corresponding to dimeric DrOTOP1 in lipid nanodisc were pooled and concentrated to 2 – 5 mg/mL with a 100 kDa cutoff centrifugal concentrator (Amicon® Ultra or Pierce™). For the DrOTOP1_C11 final cryo-EM sample, 30 μL of DrOTOP1 at 2 mg/mL was mixed 0.3 μL of C11 master stock (111 mM, 100% DMSO) and incubated for 2 h, to achieve a final concentration of 1.11 mM C11 and 1% DMSO. For the DrOTOP1_C2.2 sample, 40 μL of DrOTOP1 at 3.5 mg/mL was mixed with 0.2 μL of C2.2 master stock (2.17 M, 100% DMSO) and incubated for 2 hours, to achieve a final concentration of ≤ 10.85 mM C2.2 and 0.5% DMSO. Upon mixing, visible precipitation formed in the solution, likely due to C2.2 being insoluble in aqueous buffer at such high concentrations, so the sample was centrifuged at $21,100 \times g$ to remove insoluble debris, and thus the exact final concentration of C2.2 is unknown. For the DrOTOP1_C2.36 sample, 35 μL of DrOTOP1 at 5.3 mg/mL was mixed with 1.75 μL of C2.36 master stock (20 mM, 100% DMSO) and incubated for 3 h, to achieve a final concentration of 1 mM C2.36 and 5% DMSO.

## Cryo-EM sample preparation

All cryo-EM sample grids were prepared using a Vitrobot Mark IV (ThermoFisher), operating at 10 °C and 100% humidity. 3.5 μL of the final samples of DrOTOP1 small molecule complexes were applied to plasma-cleaned UltrAuFoil 1.2/1.3 300 mesh grids. Grids were blotted once using Whatman 1 filter paper, with a wait time of 10 s and blot time of 3 s for the DrOTOP1_C11 sample, a wait time of 5 s and blot time of 5 s for the DrOTOP1_C2.2 sample, and a wait time of 5 s and blot time of 3 – 4 s for the DrOTOP1_C2.36 samples. After blotting, grids were immediately plunge-frozen in nitrogen-cooled liquid ethane.

## Cryo-EM data collection

For the DrOTOP1_C2.2 sample, images were collected on a Titan Krios (ThermoFisher) microscope operating at 300 kV coupled with a K3 Summit (Gatan) direct electron detector (DED) at a nominal magnification of × 105,000, with a pixel size of 0.833 Å. One hundred sixteen frames of 30 ms exposure time were collected per movie, giving a total accumulated dose of 49.90 e$^-$/Å$^2$. Automated image collection was performed using Leginon[84] with a nominal defocus range of −0.7 to −1.8 μm and movies were aligned and dose weighted by MotionCor2[85]. In total, 9435 micrographs were collected from one grid.

For the DrOTOP1_C11 sample, images were collected on a Talos Arctica (ThermoFisher) microscope operating at 200 kV coupled with a K2 Summit (Gatan) DED at a nominal magnification of × 36,000, with a pixel size of 1.150 Å. Forty-nine frames of 200 ms exposure time were collected per movie, giving a total accumulated dose of 50.37 e$^-$/Å$^2$. Automated image collection was performed using Leginon[84] with a nominal defocus range of −0.8 to −1.8 μm and movies were aligned and dose weighted by MotionCor2[85]. In total, 3552 micrographs were collected from one grid.

For the DrOTOP1_C2.36 sample, images were collected on a Glacios (ThermoFisher) microscope operating at 200 kV coupled with a Falcon 4i (ThermoFisher) DED at a nominal magnification of × 190,000, with a pixel size of 0.725 Å and a total accumulated dose of

~ 40-50 e$^-$/Å$^2$. Automated image collection was performed using EPU 2 software (ThermoFisher) with a nominal defocus range of −0.8 to −1.8 μm, and movies were aligned and dose weighted by CryoSPARC Live[86,87]. In total, 26,378 micrographs were collected over three grids on three separate collections.

## Cryo-EM data processing

Motion-corrected and dose-weighted micrographs were imported to CryoSPARC[86] v4.4.1, v3.3.0, and v4.1.1 for cryo-EM data processing for DrOTOP1_C2.2, DrOTOP1_C11, and DrOTOP1_C2.36 datasets, respectively. The data processing pipelines are summarized as flow charts in Supplementary Fig. 10 (DrOTOP1_C2.2) and Supplementary Fig. 11 (DrOTOP1_C11 and DrOTOP1_C2.36).

For DrOTOP1_C2.2, contrast transfer function (CTF) values were estimated using CTFFIND4[88] and micrographs with CTF estimates worse than 6 Å were discarded. From the remaining 8907 micrographs, Blob Picker (80-150 Å diameter blobs) was used to pick particles, and after exclusion of junk picks, 6,147,309 particles were extracted using a box size 200 pixels (px). The particle stack was cleaned through three rounds of 2D classification, resulting in 870,311 nanodisc particles, which were then re-extracted (box size 200 px) using aligned shifts. This particle stack was subjected to three rounds of heterogeneous classification, with one nanodisc class and four junk classes, and particles from the nanodisc class were subjected to Ab-initio (k = 3) reconstruction. The volumes from this ab initio job were used to further classify the particle stack through two rounds of heterogeneous refinement. The best class containing 155,580 particles were used in a NU refinement job (CTF-on) to obtain the final C2 symmetry map at 3.14 Å resolution (CryoSPARC estimate).

For DrOTOP1_C11, CTF values were estimated using Gctf[89] and micrographs with CTF estimates worse than 5 Å were discarded. From the remaining 3364 micrographs, templates (from previous datasets) were used to pick particles, and after exclusion of junk picks, 3,410,686 particles were extracted. Skipping 2D classification, particles were then classified into nanodisc and junk particles through two rounds of heterogeneous refinement, first with six good nanodisc volumes and then with one good nanodisc volume and five junk-class volumes (from previous datasets) as input. The nanodisc class particles were pooled and re-extracted using aligned shifts, resulting in 560,838 DrOTOP1 nanodisc particles. This particle stack was classified further through Ab-initio (k = 3) reconstruction. The best class of 259,621 particles was refined using Non-uniform (NU) refinement[90] in C2 symmetry, followed by two more rounds of NU refinement in C2 symmetry, with the on-the-fly CTF refinement[91] option toggled on (CTF-on), resulting in a 3.56 Å resolution map. The particle stack was run through a 3D Variability[92] using a focused mask covering only the C-domain. A 3D variability display job (k = 5) was used to cluster the particles, and the particle stack that contained the most density in the C-domain pocket were pooled, totaling 164,766 particles, and refined using NU refinement (C2, CTF-on) to obtain the final map at 3.64 Å resolution (CryoSPARC estimate).

For DrOTOP1_C2.36, CTF values were estimated through the CryoSPARC Live pipeline and micrographs with CTF estimates worse than 5 Å were discarded. Initially, three paths of data processing were carried out in parallel from three separate data collection sessions, consisting of 8183 micrographs, 7309 micrographs and 9022 micrographs. For all paths, blob picker (80-200 Å diameter blobs) was used to pick particles, and after exclusion of junk picks, particle stacks of 2,219,595 particles, 1,822,146 particles and 3,194,273 particles were extracted, respectively. Each particle stack was cleaned through three rounds heterogeneous refinement, first round with 5 nanodisc volumes as input and the next two rounds with one nanodisc and four junk volumes as input. The nanodisc class particles from each path were subjected to two rounds of NU refinement, first in C1 symmetry and then in C2 symmetry, before being re-extracted from using aligned

shifts. The re-extracted particles were then pooled, 763,494 particles in total, and were reconstructed using an NU refinement (C2, CTF-off) to give a 3.61 Å resolution map (cryoSPARC estimate). From here, the particle stack was subjected to eight iterative rounds of classification using two parallel Ab-initio reconstructions (k = 3) from which the best class particles were pooled, duplicates removed, and reconstructed using NU refinement (C2, CTF-on), resulting in a stack of 339,402 particles that reconstructed a 3.40 Å resolution map. From here, five more iterative rounds of Ab-initio reconstruction (k = 2) classification and NU refinement (C2, CTF-on) were performed, resulting in the final stack of 290,491 particles that reconstruct the final map at 3.37 Å resolution (CryoSPARC estimate).

For all final cryo-EM maps, the gold-standard (GS) Fourier shell correlation (FSC) resolution estimates we report throughout this study are 3.23 Å, 3.73 Å, and 3.42 Å for DrOTOP1_C2.2, DrOTOP1_C11, and DrOTOP1_C2.36, respectively, which were obtained using the Remote 3DFSC Processing server[93] for a more accurate estimation of resolution, which also generated the related 3D FSC graphs. Local resolution estimation jobs were run in CryoSPARC with the corresponding final reconstruction masks (including nanodisc) to obtain local resolution maps.

## Model building, refinement, and visualization

For all structures, the DrOTOP1_apo model (PDB:6NF4) was used as the starting model. Initial building of the model into each map was performed in Coot v0.9.8.7[94,95]. Residues corresponding to poor areas of each map were unmodelled or deleted from the initial model. Restraints for CHS, cholesterol, C11, C2.2, and C2.36 were generated using eLBOW[96] in Phenix v1.20.1[97,98], with their SMILES as input and the eLBOW AM1 geometry optimization option. Iterative rounds of model refinement were performed using Coot v0.9.8.7, Phenix v.1.20.1, and Rosetta v2022.11[99]. For DrOTOP1_C11, the final model includes residues 47−109, 123−149, 161−214, 250−277, 306−326, 339−364, 380−441 and 511−583. For DrOTOP1_C2.2, the final model includes residues 47−110, 124−153, 158−216, 252−285, 305−327, 334−373, 376−443, 511−585. For DrOTOP1_C2.36, the final model includes residues 46−109, 123−147, 157−214, 255−283, 302−328, 335−367, 379−443, 511−585. The fittings of TM helices of each final model into their respective map densities are shown in Supplementary Fig. 12A−C. In addition, the density assignment of ligands, cholesterols, and CHS (Supplementary Fig. 12D−F) and the fit of cholesterols and CHS into their respective densities are shown as well (Supplementary Fig. 12G−I). When deciding between cholesterol and CHS for density assignment, we followed the model of DrOTOP1_apo (PDB:6NF4) for reference and for any extra density we modeled into, we chose whichever fit better. Comprehensive validation and EMRinger[100] score calculation were performed through Phenix to validate the final models against their respective maps. All cryo-EM structure figures were made in PyMOL v2.5 and UCSF ChimeraX v1.10.

## Quantification, statistical analysis and figure making

All data are presented as mean ± SEM unless otherwise noted, and each data point represents an independent replicant. Inhibition of compounds was quantified as $((I_{2peak}-I_1)/(I_{2peak}))*100$, where $I_1$ is the current immediately before wash-off and $I_{2peak}$ is the maximum recovered current (Supplementary Fig. 2). Hill coefficients were determined by the following equation: $Y = Bmax*X^h/(Kd^h + X^h)$. Absolute $IC_{50}$ was determined by the following equation: $Fifty = (Top + Baseline)/2$ $Y = Bottom + (Top-Bottom)/(1 + ((Top-Bottom)/(Fifty-Bottom) − 1)$ $*(AbsoluteIC_{50}/X)^{HillSlope})$. $pH_{50}$ curves (Fig. 6G inset) were determined using the following equation: $Y = Bottom + (Top-Bottom)/(1 + (EC/X)^{HillSlope})$ where $EC = EC50$ control. Graphs with electrophysiological data were made in Graphpad Prism 10 (Graphpad Software Inc). DrOTOP1 and mutant current traces were acquired with pClamp and were decimated 10-fold before exporting into Origin 6.1

**Table 2 | Data collection, processing, model refinement and validation**

| | DrOTOP1_C2.2 | DrOTOP1_C11 | DrOTOP_C2.36 |
|---|---|---|---|
| **Data collection and processing** | | | |
| Magnification | 105,000 | 29,000 | 190,000 |
| Voltage (kV) | 300 | 200 | 200 |
| Electron exposure (e–/Å²) | 49.90 | 50.37 | ~ 40–50 |
| Defocus range (µm) | –0.7 to –1.5 | –0.8 to –1.8 | –0.8 to –1.8 |
| Pixel size (Å) | 0.833 | 1.150 | 0.725 |
| Initial particle images (no.) | 6,147,309 | 3,410,686 | 7,236,014 |
| Symmetry imposed | C2 | C2 | C2 |
| Final particle images (no.) | 155,580 | 164,766 | 290,491 |
| Map resolution (Å) | 3.23 | 3.73 | 3.42 |
| FSC threshold | 0.143 | 0.143 | 0.143 |
| Map sharpening B factor (Å²) | –150.6 | –102.2 | –104.9 |
| **Model** | | | |
| Composition | | | |
| Peptide chains | 2 | 2 | 2 |
| Protein residues | 782 | 708 | 734 |
| Ligands | 14 | 8 | 12 |
| R.m.s. deviations | | | |
| Bond lengths (Å) | 0.006 | 0.006 | 0.007 |
| Bond angles (°) | 0.971 | 1.007 | 1.011 |
| Validation | | | |
| MolProbity score | 0.56 | 0.76 | 0.85 |
| Clashscore | 0.15 | 0.85 | 1.30 |
| EMRinger score | 2.39 | 1.9 | 2.13 |
| Poor rotamers (%) | 0.00 | 0.65 | 0.00 |
| Ramachandran plot | | | |
| Favored (%) | 99.73 | 98.52 | 99.42 |
| Allowed (%) | 0.27 | 1.48 | 0.58 |
| Disallowed (%) | 0.00 | 0.00 | 0.00 |
| Deposition ID | | | |
| EMDB | EMD-48227 | EMD-48234 | EMD-48235 |
| PDB | 9MFF | 9MFL | 9MFM |

(Microcal) and Coreldraw 2019 (Corel). We measured currents of all DrOTOP1 mutants (Supplementary Fig. 13A) in response to an 8 s pH 5.5 stimulus and excluded mutants with currents under 25 pA from further analysis.

Statistical calculations for all cryo-EM data processing and analysis were done in CryoSPARC and Phenix and are reported in Table 2. Statistical analysis in all other cases were performed in Graphpad Prism 10. Ordinary one-way ANOVAs corrected for multiple comparisons with Dunnett's correction were used for statistical analyzes of inhibition of mutants (Figs. 5D, L, 6H, and Supplementary Fig. 5B, C, E, and F) and for analysis of current magnitudes from all mutants tested (Supplementary Fig. 13A). Two tailed *t* tests with Welch correction were used to compare double mutants to single mutant in Fig. 5H, N). Two tailed *t* tests were used to compare double mutants to single mutants in Supplementary Fig. 5B, C, E and F. A paired *t* test was used to compare C2.2 vs vehicle control in Fig. 5H. Statistical comparison of normalized current magnitudes (ratios), non-parametric two-tailed Mann-Whitney tests were used to compare cumulative ranks. $*P < 0.05$, $**P < 0.01$, $***P < 0.001$, $****P < 0.0001$ for all statistical tests used in this study.

**Multiple sequence alignment for sequence conservation**

For the sequence alignment chart used to generate Supplementary Fig. 4, sequences of Otop1 from zebrafish (*Danio rerio*) (Uniprot ID:

Q7ZWK8), human (*Homo sapiens*) (Uniprot ID: Q7RTM1), and mouse (*Mus musculus*) (Uniprot ID: Q80VM9) were aligned using the T-COFFEE Multiple Sequence Alignment (MSA) webserver at EMBL-EBI[101] and were visualized using ESPript 3.0, using the ENDscript 2 webserver[102]. The secondary structure information was obtained from our cryo-EM structure of DrOTOP1_C2.2 and labeled accordingly.

For the sequence identity histogram data used to generate Supplementary Fig. 6B, a total of 58 sequences of Otop1, Otop2, and Otop3 proteins from various vertebrates – human (*Homo sapiens*), mouse (*Mus musculus*), chicken (*Gallus gallus*), pigeon (*Columba livia*), zebrafish (*Danio rerio*), Japanese rice fish (*Oryzias latipes*), western clawed frog (*Xenopus tropicalis*), *Microcaecilia unicolor*, saltwater crocodile (*Crocodylus porosus*), and green anole (*Anolis carolinensis*) – were downloaded from UniProt and sequences were aligned using the T-COFFEE Multiple Sequence Alignment (MSA) webserver at EMBL-EBI[101]. The resulting alignment file was uploaded to UCSF ChimeraX, and the sequence identity histogram information was used to color the DrOTOP1_C2.2 model.

**Reporting summary**
Further information on research design is available in the Nature Portfolio Reporting Summary linked to this article.

## Data availability
The cryo-EM density maps and atomic coordinates generated in this study have been deposited to the Electron Microscopy Data Bank (EMDB) and Protein DataBank (PDB), respectively, with accession IDs EMD-48227 and PDB: 9MFF for DrOTOP1_C2.2, EMD-48234 and PDB: 9MFL for DrOTOP1_C11, and EMD-48235 and PDB: 9MFM for DrOTOP1_C2.36. The cryo-EM density maps and atomic coordinates can also be accessed through the figshare repository, at [https://doi.org/10.6084/m9.figshare.28501874]. Source data are provided in this paper as a Source Data file, which includes the source data underlying Fig. 2A, C, D, F–I; 5D, F, H, K, L, N and 6G, H, and Supplementary Figs. 2; 5B–G; 10A; 11A, G and 13A. Source data (uncropped gels) for Supplementary Figs. 10B, 11B, and 11H are included in the Supplementary Information file.

## Code availability
AutoDock Vina v1.1.2 is freely available under an Apache license with few restrictions for commercial or non-commercial use, at [https://vina.scripps.edu]. AutoSite v1.0 is available as part of AutoDockFR under the GNU LGPL v2 open-source license at [https://ccsb.scripps.edu/adfr/]. Molscrub v0.1.0 code is open source and available under the GNU GPL v3 license, on GitHub at [https://github.com/forlilab/molscrub]. Meeko v0.6.1 code is open source and available under the GNU LPLv2.1 license, on GitHub at [https://github.com/forlilab/Meeko]. AutoDock-GPU v1.6 is open source and available under the GNU GPL v2 and GNU LGPL v2.1 licenses, and its source code and documentation is available on GitHub at [https://github.com/ccsb-scripps/AutoDock-GPU]. AutoGrid4AutoGrid4AutoGrid v4.2.6 is open source and available under the GNU GPL v2 license, on GitHub at [https://github.com/ccsb-scripps/autogrid]. Molcloud [https://github.com/ccsb-scripps/autogrid]. RDKit v2021.09.3 is freely available under a Creative Commons Attribution-ShareAlike 4.0 License, at [https://www.rdkit.org]. UMAP v0.5.9 and HDBSCAN v0.8.39 are available under a BSD 3-Clause License, on GitHub at [https://github.com/lmcinnes/umap] and [https://github.com/scikit-learn-contrib/hdbscan], respectively. Molcloud v0.3.0 code is available under the MIT license, on GitHub at [https://github.com/whitead/molcloud]. Raccoon2 is available as part of MGLTools v1.5.7 at [https://ccsb.scripps.edu/mgltools/].

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

## Acknowledgements

We thank W. Anderson and W. Lessin for managing the Hazen electron microscopy facility at Scripps Research, H. Turner for assistance with data collection, and C. Bowman, L. Dong, and J.C. Ducom for assistance with computational resources. We acknowledge members of the Ward lab, the Forli lab, and the Liman lab, G. Ozorowski, G.C. Lander, Z. Liang, and J. Walker for advice, consultation, and generating mutations in DrOTOP1. This work was supported by NIH grant R01 GM069832 to S.F. and R35GM152051 to E.R.L. Molecular graphics and analyses performed with UCSF ChimeraX, developed by the Resource for Biocomputing, Visualization, and Informatics at the University of California, San Francisco, with support from National Institutes of Health R01-GM129325 and the Office of Cyber Infrastructure and Computational Biology, National Institute of Allergy and Infectious Diseases.

## Author contributions

B.B., J.P.K., E.R.L., S.F. and A.B.W. conceptualized the studies. B. B. performed docking analyses, expressed and purified proteins, prepared cryo-EM samples, collected and processed cryo-EM data, built refined atomic models, and performed sequence conservation analyses. J.P.K. performed patch-clamp electrophysiology and carried out data and statistical analyses. G.M.O. generated mutants. B.B. and J.P.K. wrote the original manuscript draft, with inputs from E.R.L., S.F. and A.B.W. All authors contributed to manuscript review and editing.

## Competing interests

The authors declare no competing interests.
