## [Transparent Peer Review file · Nature Communications]

Structure-Guided Discovery of Otopetrin 1 Inhibitors Reveals Druggable Binding Sites at the Intrasubunit Interface

Corresponding Author: Dr Andrew Ward

Version 0:

Reviewer comments:

Reviewer #1

(Remarks to the Author)

This study identifies novel inhibitors of the zebrafish Otop1 channel and explores their structural basis of inhibition. The results reveal that these compounds target the intrasubunit pocket of zfOTOP1 by displacing cholesterol. While these findings are novel and of broad interest to the field, there are issues with data presentation, gaps in the background on OTOP channel modulators, and insufficient clarification of the identified pocket's significance. My specific comments to the author are as follows:

1)The authors initially conducted virtual screening to identify potential OTOP1 modulators targeting the C-domain pocket. However, the identified compounds unexpectedly bind to the intrasubunit pockets, which undermines the rationale of the initial experimental design. The authors should present this result concisely, as the study's primary focus is the discovery of novel inhibitor binding pockets. However, the reasons for this "off-target" effect should be discussed – does it result from structural similarities between these pockets? Additionally, since a major advancement of this study is the identification of intrasubunit antagonist binding pockets, the authors are strongly encouraged to perform another round of virtual screening targeting these pockets. Identifying new inhibitors with stronger activity would further validate the significance of the proposed antagonists binding pockets. Alternatively, the authors could refine the identified compounds (C2.2, C2.36, C11) to enhance their affinities based on the 3D structure of their binding pockets.

2)The title does not adequately reflect the main findings, particularly the discovery of two potential antagonist binding pockets in the zebrafish OTOP1 channel.

3)Page 2, Line 27: The claim that OTOP channels conduct currents in response to intracellular pH changes lacks direct evidence. Currently, no studies have analyzed OTOP1 activity by directly altering intracellular pH under inside-out patch-clamp conditions. While NH_4^+ activates OTOP1 and increases intracellular pH, the relationship between channel activation and pH rise remains unclear.

4)Page 2, Line 30: The authors state that no specific inhibitors have been reported but do not assess the selectivity of their identified compounds. Such analysis is strongly recommended, especially if these compounds are to be used to investigate OTOP1's physiological and pathological roles.

5)Page 3, Line 63: The study by Li et al. does not conclude that the OTOP1 channel in *C. elegans* functions as a taste receptor. This reference should be removed.

6)Page 4, Line 79: Hughes et al. did not demonstrate that suramin directly inhibits OTOP1. While their study showed that suramin suppresses the OTOP1-dependent Ca^{2+} response in COS7 cells, direct inhibition remains unconfirmed.

7)Page 4, Lines 86–89: The authors correctly state that Cibacron Blue 3GA is a promiscuous OTOP1 inhibitor, with an affinity comparable to that of C2.2. However, they may have missed the latest study by Kong et al., which reported the discovery and functional characterization of three preferential positive allosteric modulators of mammalian OTOP1 (with EC_{50} s of 10 - 20 μM). Thus, the statement "OTOPs currently lack effective pharmacological modulators" is inaccurate.

8)Figures should be cited sequentially in the main text. Many figure citations in the current version do not follow this rule,

reducing readability. Additionally, some figures (e.g., Figure S7, Figure S2D) are not cited in the manuscript.

9)Page 6, Lines 151–153: The inhibition was quantified by analyzing the off-response upon compound removal. However, this strategy may miss active compounds if they inhibit OTOX1 slowly (at a rate similar to or slower than current decay) and irreversibly. This possibility should be discussed.

10)Page 5, Line 129: The meaning of "n.d." in ("Lead-like & Drug-like Compounds," n.d.) is unclear. The same issue appears in Lines 163, 178, 179, 418, and 429.

11)Page 8, Lines 203–208: MolGpka (<https://xundrug.cn/molgpka>) predicts that the protonation state of C2.2 may change with pH. The authors should discuss whether protonated and deprotonated C2.2 inhibit the channel with different affinities, as suggested by the observations in Figure 2H.

12)Page 8, Line 227: The manuscript (Lines 212–214) states that no electrostatic interactions exist between the compound and the channel. Why is "electrostatics" mentioned here?

13)The I565W mutation in the C-domain pocket significantly reduces the apparent affinity of C2.2. Is this due to long-range conformational coupling, similar to how the AGAP/W38F peptide toxin inhibits the HV1 channel?

14)Page 9, Lines 255–257: The reasoning is unclear. It seems the authors initially attempted Cryo-EM structure analysis with the low-affinity compound C11 before testing higher-affinity compounds. Why?

15)Page 11, Lines 312–313: The authors claim that similar conformational changes were observed in the zfOTOX1_C11 structure and the zfOTOX1_C2.2 complex structure. How is this possible if C11 does not bind to the pocket?

16)Page 12, Lines 340–343: Please provide references.

17)Page 13, Lines 351–352: The reported Hill coefficient change (~0.3) is not significant enough to conclude reduced positive cooperativity.

18)The authors fail to identify a single critical residue in the central pocket greatly affecting C2.2 binding (the E384A mutation shows limited effect), despite Cryo-EM data suggesting compound binding. Why?

19)Page 14, Line 391: Incorrect figure citation—please correct it. Please review the whole manuscript to avoid similar mistakes.

20)Page 15, Line 423: The phrase "a 6.67 μ M inhibitor" is not presented scientifically. Please rephrase.

21)Page 15, Lines 429–432: Experimental data are needed to support the claim that all identified compounds share the same binding site. Testing these compounds on the G130W mutant (which most strongly affects C2.2 binding) is recommended.

22)Page 16, Lines 450–451: The A388 mutation does not significantly contribute to the reduced C2.2 binding affinity; the 50-fold reduction is primarily due to the G130W mutation. Reword for clarity.

Reviewer #2

(Remarks to the Author)

Otop channels have recently been identified as proton channels that play key roles in sour taste and other physiological functions. Identifying specific small-molecule ligands for Otop channels will facilitate functional studies of this new class of channels and may have potential applications in health and food sciences. The current manuscript by Burendei et al. identifies a group of inhibitors for the zebrafish Otop1 channel through high-throughput virtual screening, cryo-EM structure determination, and electrophysiology experiments. Their approaches are novel and the findings are very interesting. The binding sites of these molecules identified by cryo-EM structure may also help to understand the gating of this channel and conductance of protons. The reviewer finds these results exciting overall, although some data quality could be improved, and certain conclusions drawn from the structural findings may be questionable due to missing key evidence.

1. Since the cryo-EM structures show different binding sites for these molecules compared to the docking results, would the authors be able to identify the sites found in cryo-EM if they performed another round of docking with C11, C2.2, and C2.36 while expanding the search box? If so, this would further validate their cryo-EM structures.

2. It would be helpful to show the structures of C11 and C2.2 in Fig. 2A and B.

3. Why does C11 have a mild effect on human and mouse Otop1, while C2.2 has no effect on them, if they bind at the same sites?

4. Lines 152–153: "...inhibition was measured by analyzing the off response upon removal of the compound." A clearer explanation of how this calculation was performed is needed.

5. It is necessary to include current traces with 0 μ M inhibitor in Fig. 2C and Fig. 5H–K.

6. Fig. 2C shows that higher concentrations of C11 lead to a larger "tail current" after treatment. What could be the reason for

this? Does the treatment slow down inactivation, or is there some form of potentiation after inhibitor washout, similar to Zn²⁺ blockade observed in other Otop channels?

7. Since C11 was not observed in the cryo-EM structure and the current mutational evidence (Fig. 5D, none of the tested mutations affected C11's inhibition) does not support its binding at the same site as C2.2, the authors should acknowledge the possibility that C11 inhibits the channel by binding to a different site.

8. The densities for C2.36 in Fig. 4C and D appear to be relatively weak. The current evidence does not fully support the proposed model, especially for the two C2.36 molecules in the central site. Could these densities correspond to molecules other than C2.36?

9. Only the G130W mutation was tested for C2.36 in Fig. 5F. Have the effects of other mutations, particularly those in the central site tested in Fig. 5D and E, been examined?

10. To enhance the effects of mutations, the authors introduced the double mutations G130W/A388W and G130W/A274W. However, neither A388W nor A274W had any effect individually (Fig. 5D). What was the rationale for combining these mutations with G130W, given that they are not in the same binding site? Additionally, how can the larger effect observed in the G130W/A388W double mutant be explained, considering A388W alone had no effect? Meanwhile, since the E384A mutation in the central site already affected C2.2 inhibition, why not test the G130W/E384A double mutant for both C11 and C2.2?

11. How conserved are these binding sites in humans and mice? A sequence alignment between these species, highlighting whether the key residues are conserved, would be useful.

12. Most samples in Fig. 4D–F have only three data points, which is insufficient for a reliable t-test. The same issue applies to Fig. 6G and H.

13. In Fig. 6G, fitted curves illustrating the shift in proton sensing would improve readability.

Reviewer #3

(Remarks to the Author)

Burendei et al. present cryo-EM structures of zebrafish OTO1 (zfOTO1) in complex with small molecule inhibitors and characterize their binding sites through a combination of virtual screening, functional testing, cryo-EM, and mutagenesis. The authors initially aimed to identify inhibitors by virtual screening against the C-domain pocket of zfOTO1. However, cryo-EM structures of zfOTO1 with two potent inhibitors, C2.2 and C2.36, at resolutions of 3.2 Å and 3.4 Å respectively, revealed that these inhibitors bind to two unexpected sites in the intrasubunit interface, termed the "outer site" and the "central site". The study used mutagenesis experiments to confirm the importance of residues within these intrasubunit sites for inhibitor binding and function.

The manuscript presents an interesting structure-based virtual screening study targeting zfOTO1, contributing valuable insights into potential modulators of OTO channels. The cryo-EM maps are of high quality, and the work offers a solid foundation for further pharmacological exploration of this under-studied family. However, there are several points that would benefit from clarification or further elaboration to strengthen the manuscript. These are outlined below:

1) The authors are encouraged to discuss what type of modulator of zfOTO1 (and other OTOs) would be desirable from the perspective of its physiological role in humans. This would help frame the relevance of the screening approach.

2) The authors should clearly state which functional conformation of zfOTO1 was used for virtual screening and discuss how this choice might affect the initial hits and subsequent analysis. From a functional or medical standpoint, which conformational state would be most relevant to target?

3) The authors should clarify whether the two zfOTO1 monomers influence each other's function. This might be known from earlier work, but the manuscript does not currently make it clear. It would be helpful to specify whether this interaction is relevant to the *in silico* analysis or the functional assays.

4) Were any of the datasets processed extensively in C1 symmetry? If so, what insights did that provide? Also, did the authors explore symmetry expansion to classify protomers in potentially different conformations, e.g., with distinct ligand occupancies?

5) The authors are encouraged to emphasize sequence similarities and identity among the resolved OTO structures, particularly between zfOTO1 and mammalian orthologs. This would help contextualize the relevance of using zfOTO1 for virtual screening, especially in light of Fig. S10, which could be referenced earlier in the results section.

6) Line 136: The term "visual inspection" is vague. Could the authors clarify what criteria were used during this process? Were there specific features being assessed? For example, were particular structural features, docking poses, or scoring thresholds taken into account?

7) The similarity between the identified scaffold and that of tricyclic antidepressants (TCAs) is noteworthy. Are TCAs or structurally related compounds known to modulate sour taste in humans? A brief reflection on this point would add an intriguing pharmacological dimension to the findings.

8) While the authors hypothesize that the binding of compounds such as C11 and C2.2 may not involve interactions with titratable residues, could it be that these ligands stabilize a conformation of zfOTO1 that indirectly affects titratable residues? The authors are encouraged to reflect on this possibility, as it may influence the interpretation of their binding

mechanism.

9) The manuscript uses CHS and CHOL somewhat interchangeably in describing bound lipid molecules. Given the similar density features of these sterols in cryo-EM maps, how were they distinguished in the models? If the presence of CHS and/or CHOL was confirmed experimentally, for instance by mass spectrometry, this should be explicitly mentioned in the text.

Version 1:

Reviewer comments:

Reviewer #1

(Remarks to the Author)

The authors have adequately addressed all of my concerns. I recommend this paper for publication.

Reviewer #2

(Remarks to the Author)

The authors have addressed most of my concerns.

For their response to my comment #1, rather than performing blind docking, would the authors consider conducting new docking studies using the current molecules specifically on the binding pockets identified in the Cryo-EM structures? This approach would allow for a focused evaluation of the experimentally determined sites. It is particularly valuable for the binding pockets that in the case where two C2.36 molecules were assigned to the central site, where the cryo-EM densities are not very clear. In their response to my comment #8, the authors also mention that they are uncertain about the assignment of C2.36 at that site, particularly the second copy.

Also, since the Cryo-EM structures have shown that the initial docking sites were incorrect, I believe it would be more appropriate to move the docking results currently shown in Figure 1 to the supplementary information, rather than including them in the main text.

Reviewer #4

(Remarks to the Author)

This is a compelling manuscript reporting the discovery of several new modulators of OTOP1. The data is of excellent quality and well-presented. I was brought on to the review in place of reviewer #3. I find that the authors have addressed the major concerns of reviewer #3, and I congratulate the authors on this work.

Response to reviewers:

REVIEWER COMMENTS

Reviewer #1 (Remarks to the Author):

This study identifies novel inhibitors of the zebrafish Otop1 channel and explores their structural basis of inhibition. The results reveal that these compounds target the intrasubunit pocket of zfOTOP1 by displacing cholesterol. While these findings are novel and of broad interest to the field, there are issues with data presentation, gaps in the background on OTO channel modulators, and insufficient clarification of the identified pocket's significance. My specific comments to the author are as follows:

1)The authors initially conducted virtual screening to identify potential OTO1 modulators targeting the C-domain pocket. However, the identified compounds unexpectedly bind to the intrasubunit pockets, which undermines the rationale of the initial experimental design. The authors should present this result concisely, as the study's primary focus is the discovery of novel inhibitor binding pockets. However, the reasons for this "off-target" effect should be discussed —does it result from structural similarities between these pockets? Additionally, since a major advancement of this study is the identification of intrasubunit antagonist binding pockets, the authors are strongly encouraged to perform another round of virtual screening targeting these pockets. Identifying new inhibitors with stronger activity would further validate the significance of the proposed antagonists binding pockets. Alternatively, the authors could refine the identified compounds (C2.2, C2.36, C11) to enhance their affinities based on the 3D structure of their binding pockets.

We thank the reviewer for their comments regarding the identification of unexpected binding pockets. Comparing the new binding sites (intrasubunit outer site and intrasubunit central site) to the original targeted pocket (C-domain pocket), the C-domain pocket is relatively small and is more hydrophilic, as it has many hydrogen bond donors and acceptors present, whereas the intrasubunit sites are facing either the lipid bilayer or the central cavity filled with lipids, are larger in size, and more hydrophobic in nature. Specifically for the outer site, the induced-fit of inhibitor binding makes the pocket larger and more accommodating for small molecules as sidechains of M133 and M573 shift to make space, compared to its apo conformation. Within this context, we believe that the inhibitors favored the intrasubunit sites due to a combination of the

41 following circumstances: 1) Small molecules that were docked favorably to the C-
42 domain pocket were smaller and had lower molecular weights. As such, these smaller
molecules can also occupy larger pockets with minimal restrictions. 2) The Chembridge
library we screened consists of drug-like molecules which tend to be moderately
lipophilic. Although we selected and tested the best molecules for a hydrophilic pocket,
the selected molecules were still relatively lipophilic. Because of these circumstances,
we think the molecules were still lipophilic enough to occupy the hydrophobic
intrasubunit sites quite favorably and that functional screening identified the molecules
that then caused functional effects. Moreover, the fundamental limitation of ignoring the
induced fit during the docking might have contributed to the inaccurate predictions. We
have added a concise version of the above explanation to the Discussion section (lines
467-473).

We agree that a subsequent round of virtual screening focusing on the
intrasubunit sites is the best next step to identify more active molecules against
zebrafish OTOA1. However, we believe such a screen is outside of the scope of this
current work, whose major contribution as the reviewer appreciates is the discovery of
an antagonist binding pocket. Similarly, we believe that refining of the current best
inhibitors, which will likely be a major endeavor encompassing rational design and
potential chemical synthesis of small molecules, is a reasonable next step but outside
the current scope; moreover, given that molecules identified in this screen are specific
for zebrafish OTOA1 and not likely to be of wide-spread utility for physiological or
clinical uses, such a screen is not warranted.

We further agree that some of the text could be tightened to make it more
concise, and we have endeavored to do so by revising the text throughout the
manuscript.

*2)The title does not adequately reflect the main findings, particularly the discovery of*
*two potential antagonist binding pockets in the zebrafish OTOA1 channel.*

We have modified the title to reflect our findings more accurately.

*3)Page 2, Line 27: The claim that OTOA channels conduct currents in response to*
*intracellular pH changes lacks direct evidence. Currently, no studies have analyzed*
*OTOA activity by directly altering intracellular pH under inside-out patch-clamp*
*conditions. While NH₄⁺ activates OTOA and increases intracellular pH, the relationship*
*between channel activation and pH rise remains unclear.*

This line in the abstract was removed and we replaced it by now simply
mentioning that OTOA contributes to sour and ammonium tastes, a claim backed by
direct evidence.

4)Page 2, Line 30: The authors state that no specific inhibitors have been reported but do not assess the selectivity of their identified compounds. Such analysis is strongly recommended, especially if these compounds are to be used to investigate OTOP1's physiological and pathological roles.

We removed the line, which might imply that we have identified specific inhibitors of OTOP1 channels. We agree that we did not demonstrate that C2.2 (or C11 or C36) are specific for OTOP1, although unlike Cibacron blue and Zn²⁺ they do not have known targets. We measured specificity of these compounds only in comparison to two closely related proteins – mouse and human OTOP1, which are more closely related to zebrafish OTOP1 than any other protein in zebrafish. This was an important control, as it shows that the effects of C2.2 (and C11) are not non-specific (e.g. by changing pH, or acting in a very promiscuous manner). We chose not to look at specificity with respect to a large panel of ion channels, as the compounds we identified could only be used in zebrafish, and therefore are unlikely to be of utility for investigating OTOP1's physiological or pathological roles in mammals. Instead, our work, by identifying this critical modulatory site now sets the stage for future screens to identify specific modulators of mouse and human OTOP channels.

5)Page 3, Line 63: The study by Li et al. does not conclude that the OTOP1 channel in C. elegans functions as a taste receptor. This reference should be removed.

Thank you for noticing this. We replaced with the correct reference which is Mi et al, 2021.

6)Page 4, Line 79: Hughes et al. did not demonstrate that suramin directly inhibits OTOP1. While their study showed that suramin suppresses the OTOP1-dependent Ca²⁺ response in COS7 cells, direct inhibition remains unconfirmed.

We thank the reviewer for pointing out this mistake and have modified this sentence in the manuscript to be more accurate.

7)Page 4, Lines 86–89: The authors correctly state that Cibacron Blue 3GA is a promiscuous OTOP1 inhibitor, with an affinity comparable to that of C2.2. However, they may have missed the latest study by Kong et al., which reported the discovery and functional characterization of three preferential positive allosteric modulators of mammalian OTOP1 (with EC50s of 10 - 20 μM). Thus, the statement “OTOPs currently lack effective pharmacological modulators” is inaccurate.

This study was published after we had submitted our manuscript. We have
added the reference to our introduction text and modified it to reflect the latest findings
in the field. (lines 79 - 80)

*8) Figures should be cited sequentially in the main text. Many figure citations in the*
*current version do not follow this rule, reducing readability. Additionally, some figures*
*(e.g., Figure S7, Figure S2D) are not cited in the manuscript.*

We thank the reviewer for pointing this out. We have revised our manuscript and
figures such that our figures are now in sequential order. We used “see below”
wherever applicable, as per the Nature Communication formatting guidelines. We have
also included references to all uncited figure panels throughout our manuscript.

*9) Page 6, Lines 151–153: The inhibition was quantified by analyzing the off-response*
*upon compound removal. However, this strategy may miss active compounds if they*
*inhibit OTOPI slowly (at a rate similar to or slower than current decay) and irreversibly.*
*This possibility should be discussed.*

This is true and the analysis was a bit more nuanced than described. The off-
response method we employed allows us to quantitatively screen through compounds
and mutants and is very reliable and robust. But in addition to this method, we used
“visual inspection” to look for compounds that inhibited the channels that might be
poorly reversible, which is how we identified C2.36. This visual inspection, which is now
mentioned in the text (lines 146 - 148) involved looking for a change in the decay rate
after introduction of the chemical and also looking for slow recovery of the currents. This
was hard to quantify, and we found it was best done by eye. It should also be noted that
we do not claim that the compounds that we classified as having little inhibitory activity
could not under some circumstances (e.g. longer incubation or incubation at a different
pH) be effective inhibitors.

*10) Page 5, Line 129: The meaning of “n.d.” in (“Lead-like & Drug-like Compounds,”*
*n.d.) is unclear. The same issue appears in Lines 163, 178, 179, 418, and 429.*

“n.d.” stands for “no-date”. This results from the reference manager software
adding citations to entries that have no date associated with them, e.g. website links,
github links. We have changed our citation and reference formatting to match the
guidelines and “n.d.” is no longer present in the manuscript.

*11) Page 8, Lines 203–208: MolGpka (<https://xundrug.cn/molgpka>) predicts that the*
*protonation state of C2.2 may change with pH. The authors should discuss whether*

*protonated and deprotonated C2.2 inhibit the channel with different affinities, as*
*suggested by the observations in Figure 2H.*

We thank the reviewer for raising this point. The report from MolGpka (using
SMILES: "C=C(C)COc1ccccc1NC(=O)c1nc[nH]c1C") agrees with the state of C2.2 we
presented in Figure 5A, B, and based on the predicted pKa values, the protonation state
of C2.2 should not change between pH 4.0 - 11.3. As our experimental pH conditions
are within this range (lowest tested pH is 5.0 and highest is pH 7.4), we expect that
protonation and deprotonation of C2.2 are not relevant factors in our experiments.

The data in Figure 2H, I shows that there is little to no effect of pH on inhibition of
zebrafish OTOP1 by C11 or C2.2, consistent with the interpretation that these
compounds do not have groups that are titratable at the pH values we tested. The small
shift in the IC50 for C2.2 between pH 5.5 and pH 5.0 could either be attributable to
titration of residues in the binding sites or changes in channel gating that could affect
the ability of the compound to reach its binding site, or could otherwise affect its ability
to inhibit the channel (e.g. if it acts as a gating modifier). In the outer site, the key
protonatable residues are E267 and H574. We believe that E267 is protonated in our
structures, even at neutral pH, as the tool PROPKA 3
(<https://github.com/jensengroup/propka>) predicts its pKa to be ~9. However, for H574,
the predicted pKa is ~5. If H574 is protonated, it could clash with the protonation
present on C2.2 on the heterocycle, in which case any potential interaction could be
weakened. We have added our thoughts regarding this point to the discussion section
(line 476 - 483)

*12)Page 8, Line 227: The manuscript (Lines 212–214) states that no electrostatic*
*interactions exist between the compound and the channel. Why is "electrostatics"*
*mentioned here?*

Changed to remove mention of electrostatics.

*13)The I565W mutation in the C-domain pocket significantly reduces the apparent*
*affinity of C2.2. Is this due to long-range conformational coupling, similar to how the*
*AGAP/W38F peptide toxin inhibits the HV1 channel?*

We believe that this may be the case, that the I565W mutation may act at a
distance to change gating, rather than directly disrupting the ligand in its binding pocket
for the following reasons (1) our cryo-EM structure shows that the ligands do not bind in
close proximity to I565W and (2) the I565W mutation has a relatively severe effect on
gating of the channels, making them appear to be "easier to open" (and consequently
harder to inhibit). However, we do not want to overinterpret our data due to the difficulty

in studying and quantitating gating of OTO channels (see Teng et al., 2022). We do
not have direct evidence, of the type that can be obtained when studying the voltage-
gated proton channel Hv1, to support such a conclusion. We now discuss this point with
reference to the appropriate literature (lines 511-520)

*14)Page 9, Lines 255–257: The reasoning is unclear. It seems the authors initially*
*attempted Cryo-EM structure analysis with the low-affinity compound C11 before testing*
*higher-affinity compounds. Why?*

We thank the reviewer for pointing out this ambiguity. This is because we found
C11 earlier in the study and determined the DrOTOP1_C11 cryo-EM structure before
we identified C2.2 and C2.36 as better inhibitors. We presented our data slightly out of
chronological order in an effort to be more succinct.

*15)Page 11, Lines 312–313: The authors claim that similar conformational changes*
*were observed in the zfOTOP1_C11 structure and the zfOTOP1_C2.2 complex*
*structure. How is this possible if C11 does not bind to the pocket?*

We thank the reviewer for catching this and we rephrased the sentence to be
clearer. Our intention was to indicate that simply no large-scale conformational changes
were observed in any of our cryo-EM structures.

Here, we note that we did not see clear evidence of C11 binding in any of the
pockets we considered, and only observed small local changes in the intrasubunit
interface region (Supplementary Figure S7C). A possible explanation is that C11 binds
to the intrasubunit sites like C2.2 and C2.36 and causes a similar local conformational
change, albeit not to the same degree, but its binding could not be captured by cryo-EM
or as easily perturbed by single residue mutations due to its lower affinity and smaller
size.

*16)Page 12, Lines 340–343: Please provide references.*

We have now referenced Supplementary Figure 4 where we identified I565W
which reduces C11 and C2.2 potency. (lines 343-346)

*17)Page 13, Lines 351–352: The reported Hill coefficient change (~0.3) is not significant*
*enough to conclude reduced positive cooperativity.*

We agree and have removed this interpretation.

*18)The authors fail to identify a single critical residue in the central pocket greatly*
*affecting C2.2 binding (the E384A mutation shows limited effect), despite Cryo-EM data*
*suggesting compound binding. Why?*

The reviewer is correct that we did not succeed in generating any mutations of
the central site that produced large effects on the activity of C2.2. This could argue that
the compound is acting solely through binding to the outer site, and that the observed
density at the central site represents non-functional binding. We would probably take
this position except that the Hill coefficient suggests that C2.2 has two binding sites.
Considering that OTOP1 is a dimer, this could mean that it needs to bind to both
subunits or it could mean that it binds to two sites on each subunit. Our observation that
we could further reduce the IC₅₀ or maximal block by C2.2 with mutations of the central
site in two double mutants, although the effects are relatively modest, makes us favor
the latter interpretation. This is now discussed in the discussion (Lines 489 - 493).

*19)Page 14, Line 391: Incorrect figure citation—please correct it. Please review the*
*whole manuscript to avoid similar mistakes.*

We thank the reviewer for pointing this out. We have corrected this and have
thoroughly checked over the entire manuscript for similar mistakes.

*20)Page 15, Line 423: The phrase “a 6.67 μM inhibitor” is not presented scientifically.*
*Please rephrase.*

We thank the reviewer for raising this point. We have rephrased the sentence to
be more accurate.

*21)Page 15, Lines 429–432: Experimental data are needed to support the claim that all*
*identified compounds share the same binding site. Testing these compounds on the*
*G130W mutant (which most strongly affects C2.2 binding) is recommended.*

We now clearly state that it's only an assumption that all identified compounds
have the same binding site. We chose not to test all the compounds against the G130W
mutant as testing against one mutant in any case would not prove (or disprove) this
contention which would need to be supported by cryo-EM structures, which are outside
the scope of the current study. (lines 447 - 448).

*22)Page 16, Lines 450–451: The A388 mutation does not significantly contribute to the*
*reduced C2.2 binding affinity; the 50-fold reduction is primarily due to the G130W*
*mutation. Reword for clarity.*

We thank the reviewer for pointing this out. The mutation does in fact shift the dose-response curve further to the right (Figure 5K) and changes the IC50 compared to G130W alone, but only by 3-fold. We have rewritten the text (lines 379 - 380).

Reviewer #2 (Remarks to the Author):

Otop channels have recently been identified as proton channels that play key roles in sour taste and other physiological functions. Identifying specific small-molecule ligands for Otop channels will facilitate functional studies of this new class of channels and may have potential applications in health and food sciences. The current manuscript by Burendei et al. identifies a group of inhibitors for the zebrafish Otop1 channel through high-throughput virtual screening, cryo-EM structure determination, and electrophysiology experiments. Their approaches are novel and the findings are very interesting. The binding sites of these molecules identified by cryo-EM structure may also help to understand the gating of this channel and conductance of protons. The reviewer finds these results exciting overall, although some data quality could be improved, and certain conclusions drawn from the structural findings may be questionable due to missing key evidence.

1. Since the cryo-EM structures show different binding sites for these molecules compared to the docking results, would the authors be able to identify the sites found in cryo-EM if they performed another round of docking with C11, C2.2, and C2.36 while expanding the search box? If so, this would further validate their cryo-EM structures.

We thank the reviewer for the suggestion. The experiment suggested would be a blind docking of the inhibitors to the entire zfOTOP1 protein. However, blind docking is generally considered less accurate than docking in a specific pocket. This is due to the larger search space which is difficult to fully sample, which results in more false positive docking results as the ligand gets stuck in local energy minima. As such, even if blind docking were able to place the inhibitors in the intrasubunit site, we suspect it is just as likely to place it in another area of the protein with a similar score, and thus the results would not be very meaningful. Moreover, we do not model any induced fit effects during docking and this could further reduce the accuracy of the predictions, as different areas of the protein would “mold” to ligand binding differently, further emphasizing the importance of sticking to a single pocket when interpreting docking results and comparing docking scores.

As we have mentioned in the manuscript, we consider the intrasubunit interface
sites to double as reasonable binding sites for exogenous small molecules as they have
evolved to bind cholesterol.

*2. It would be helpful to show the structures of C11 and C2.2 in Fig. 2A and B.*

We thank the reviewer for this suggestion and have included chemical structures
of C11 and C2.2 into Figure 2.

*3. Why does C11 have a mild effect on human and mouse Otop1, while C2.2 has no
effect on them, if they bind at the same sites?*

We thank the reviewer for this question. Here, we suspect that the answer lies in
the chemical structure differences between C11 and C2.2 and differences in the
sequences of zebrafish Otop1 and mammalian Otop1 for residues in the intrasubunit
pockets. Since C2.2 is relatively larger than C11, we speculate that the intrasubunit
pockets of mammalian Otop1 do not accommodate for molecules larger than C11.
Focusing on the intrasubunit outer site, there are a few differences in residues, such as
G130 and L134 in zfOtop1 corresponding to S136 and F140 in human Otop1. The
larger side chains of these residues in mammalian Otop1 could hinder binding of
larger inhibitors like C2.2, but still allow for binding of a smaller inhibitor like C11. Similar
differences in the central site are also present, with A274 and L567 in zfOtop1
corresponding to T283 and F593 in human Otop1, where mammalian Otop1 harbors
residues with larger side chains.

However, we note that we do not fully understand the mechanism of inhibition
upon binding of inhibitors in the intrasubunit binding sites, ultimately, a bound-pose of
C11 in the intrasubunit sites of zfOtop1 or cryo-EM structures of mammalian Otop1
might be necessary to explain why exactly C11 has a mild effect on mammalian Otop1
and C2.2 does not.

We also note that C2.2 was tested at a lower concentration (100 μ M) than C11
(300 μ M), in the experiment shown in Fig. 2G. Perhaps C2.2 could also mildly inhibit
mammalian Otop1 if tested at 300 μ M.

*4. Lines 152–153: "...inhibition was measured by analyzing the off response upon
removal of the compound." A clearer explanation of how this calculation was performed
is needed.*

We thank the reviewer for requesting this clarification. We now show in
Supplementary Figure 1 how and where the measurements were made to analyze the
"off-response", or the recovered current upon chemical wash-off. We also clearly report

the equation used in the figure legend and in the methods (Lines 869 - 871). Note that
this eliminates false positives that would come from measuring the response at the end
of the chemical application to before (on response).

*5. It is necessary to include current traces with 0 μ M inhibitor in Fig. 2C and Fig. 5H–K.*

The reviewer asked quite reasonably to see the current decay in the absence of
inhibitor to compare with its presence. We now show the vehicle control (0.3% DMSO)
alongside responses to 100 μ M C11 and 10 μ M C2.2 in supplementary figure S1. This
shows the typical decay of the currents, with no recovery upon removal of the vehicle,
indicating no non-specific effects of DMSO and no artifact from the solution exchange.
We have also added a vehicle control to Figures 2A and 2D. For Figure 2C (now 2B),
we did not do this control at the time, as it the experiment involved a series of
concentrations including several that were subthreshold (3 μ M for C11). In this case,
please refer to supplementary figure 2 for vehicle controls. Additionally, we added
vehicle controls to Figs. 5G and 5M, where only a single concentration of C2.2 was
tested, to help the reader more easily see the difference between the inhibitor and
vehicle control for those mutants. In the case of Fig 5E, I, and J, C2.2 is used at a series
of concentrations including two (3 μ M or 10 μ M) that were subthreshold (so no different
from vehicle control).

*6. Fig. 2C shows that higher concentrations of C11 lead to a larger “tail current” after*
*treatment. What could be the reason for this? Does the treatment slow down*
*inactivation, or is there some form of potentiation after inhibitor washout, similar to Zn^{2+}*
*blockade observed in other Otop channels?*

In this figure it indeed appears that currents after washoff were larger for higher
concentrations of the inhibitor. However, this was not a consistent observation, and we
believe that it is instead reflects variability in the magnitude of the currents, which
tended to decay over time, such that the currents were largest for the highest
concentrations, which tended to be tested first. We did not test to see if inhibition
protected the currents from decay (say by reducing proton influx, which degrades the
ionic gradient), which could very well be the case. There was no evidence for
potentiation like we see for Zn^{2+} where the wash off currents can be larger than the
initial currents (especially true for OTOP3).

*7. Since C11 was not observed in the cryo-EM structure and the current mutational*
*evidence (Fig. 5D, none of the tested mutations affected C11’s inhibition) does not*
*support its binding at the same site as C2.2, the authors should acknowledge the*
*possibility that C11 inhibits the channel by binding to a different site.*

We thank the reviewer for the suggestion. We have included this discussion to
our text, in the discussion section (lines 496-497).

*8. The densities for C2.36 in Fig. 4C and D appear to be relatively weak. The current*
*evidence does not fully support the proposed model, especially for the two C2.36*
*molecules in the central site. Could these densities correspond to molecules other than*
*C2.36?*

We thank the reviewer for pointing this out. Here, concerning the intrasubunit
outer site, we argue that the density for C2.36 in Fig. 4C (now Fig. 4D) is comparable to
the density for C2.2 in Fig. 3E and we observe local side-chain perturbations as shown
in Fig. 5B, so we would claim that C2.36 binds to the intrasubunit outer site.

For the intrasubunit central site, shown in Fig. 4D (now Fig. 4F), we agree that
the density is more ambiguous. Fitting two copies of C2.36 in the central site was a
result of our best attempt to model C2.36 into the density in the region. While the
density could potentially correspond to cholesterol molecules that are often found in the
central cavity of OTOP channels or other lipid molecules, we decided to model the first
C2.36 copy due to the local conformational changes observed in F277 and TM6. The
modeling of the second C2.36 copy (one on top, that extends towards L395), was
simply based on its fit in the density, and we observed no side chain perturbations near
it.

We have edited our text to reflect this more accurately, in the results section
(lines 311 - 315). If the reviewer requests so, we can remove the second copy of C2.36
in the central site from the final deposited model, but we would like to keep the copy
near F277, as well as the copy in the intrasubunit outer site.

*9. Only the G130W mutation was tested for C2.36 in Fig. 5F. Have the effects of other*
*mutations, particularly those in the central site tested in Fig. 5D and E, been examined?*

We did not test mutations of the central site as C2.36 was tricky to work with,
given its poor reversibility and we therefore chose to focus on the mutation that was
likely to have the largest effect. These data are merely meant to be supportive of the
main results of the paper, which concerns the binding site for C2.2, for which we have
better structural evidence (see above).

*10. To enhance the effects of mutations, the authors introduced the double mutations*
*G130W/A388W and G130W/A274W. However, neither A388W nor A274W had any*
*effect individually (Fig. 5D). What was the rationale for combining these mutations with*
*G130W, given that they are not in the same binding site? Additionally, how can the*

*larger effect observed in the G130W/A388W double mutant be explained, considering*
*A388W alone had no effect? Meanwhile, since the E384A mutation in the central site*
*already affected C2.2 inhibition, why not test the G130W/E384A double mutant for both*
*C11 and C2.2?*

These are all excellent points, and we have now tested the double mutation with
E384A (the reason we did not test it earlier is historical). Intriguingly, we found that the
double mutation of G130W/E384A instead of shifting the IC50 like G130W/A388W,
significantly reduced the maximal inhibition by C2.2 (to around 50%). Given that this
result was completely unexpected, in parallel we re-tested G130W, which was fully
inhibited by the higher concentrations of C2.2. We did not test C11 against
G130W/E384A or E384A as the chemical is no longer available from ChemBridge and
is a less potent inhibitor, we have decided not to pursue further testing of C11. The
result support a contribution of E384 in the central site, and suggest that inhibition may
involve a complex interplay between the two sites. This is now discussion in (Lines 489
454 - 492).

Regarding the double mutation of G130W with A388W or A274W, although the
single mutations had very modest effects, we generated the double mutants to see if
this would unmask a contribution of the central site (observed as a difference between
G130W and the double mutants). The thinking was that if we broke the high sensitivity
of the channels to the blocker (in the G130W mutant), then a change in a lower affinity
site might become evident, as it did. Note that we were hesitant to create and test
double mutants for any of the single mutations that caused large changes in channel
gating (L134W, E384A, or M573W) as this could muddy the interpretation. This is now
addressed in the discussion section (Lines 506-510).

*11. How conserved are these binding sites in humans and mice? A sequence alignment*
*between these species, highlighting whether the key residues are conserved, would be*
*useful.*

We thank the reviewer for this insightful question. We presented Figure S10A
(now Supplementary Fig. 6B) as a way of conveying the sequence preservation in each
considered binding site, however, we agree that a more detailed comparison between
zfOTOP1 and mammalian OTOP1 would be useful. We have added such a sequence
alignment chart as Supplementary Figure 1.

As Fig. S10A (now Supplementary Fig. 6B) shows, the intrasubunit sites are
relatively more conserved regions of OTOP channels, so we expect these sites are
important regions related to channel gating in mammalian OTOP channels as well.
However, as we stated in response to comment 3, there are differences in some of the

less-conserved residues lining the intrasubunit sites, which could underlie differing
responses to small molecules binding in these sites.

*12. Most samples in Fig. 4D–F have only three data points, which is insufficient for a*
*reliable t-test. The same issue applies to Fig. 6G and H.*

We thank the reviewer for this comment and believe they are referring to Fig 5 D-
F (Figure 4 shows structures). For this experiment we have increased the n for all
mutants to at least 5. This confirmed our initial results and the same mutants showed
statistically significant, and relatively large changes in sensitivity to the inhibitor. For
Figure 6, all constructs tested have an $n \geq 4$. This was sufficient for the statistical
analyses performed to show gating changes in mutants as compared to WT. These
differences were also obvious from visual inspection of the traces.

*13. In Fig. 6G, fitted curves illustrating the shift in proton sensing would improve*
*readability.*

We thank the reviewer for this suggestion. We have added an inset to Figure 6G
with fitted curves for WT and each mutant.

*Reviewer #3 (Remarks to the Author):*

*Burendei et al. present cryo-EM structures of zebrafish OTO1 (zfOTO1) in complex*
*with small molecule inhibitors and characterize their binding sites through a combination*
*of virtual screening, functional testing, cryo-EM, and mutagenesis. The authors initially*
*aimed to identify inhibitors by virtual screening against the C-domain pocket of*
*zfOTO1. However, cryo-EM structures of zfOTO1 with two potent inhibitors, C2.2*
*and C2.36, at resolutions of 3.2 Å and 3.4 Å respectively, revealed that these inhibitors*
*bind to two unexpected sites in the intrasubunit interface, termed the "outer site" and the*
*"central site". The study used mutagenesis experiments to confirm the importance of*
*residues within these intrasubunit sites for inhibitor binding and function.*

*The manuscript presents an interesting structure-based virtual screening study targeting*
*zfOTO1, contributing valuable insights into potential modulators of OTO channels.*
*The cryo-EM maps are of high quality, and the work offers a solid foundation for further*
*pharmacological exploration of this under-studied family. However, there are several*
*points that would benefit from clarification or further elaboration to strengthen the*
*manuscript. These are outlined below:*

*1) The authors are encouraged to discuss what type of modulator of zfOTOP1 (and*
*other OTOPs) would be desirable from the perspective of its physiological role in*
*humans. This would help frame the relevance of the screening approach.*

We thank the reviewer for the insightful comment. We have added a sentence
(lines 80 - 83) regarding this point to our Introduction section in the manuscript. We
would like to remind the reviewer that it is still too early to know the full therapeutic
potential of OTOP channels inhibitors, so that for the short term their utility may be
mostly in determining the contribution of OTOP channels to physiology and pathology.

From a functional standpoint, we believe that while any modulator would be
exciting to identify, specific inhibitors would be the most useful. For example, inhibitors
would be useful in identifying and specifically attenuating contributions of proton
currents carried by OTOP channels in cells or tissues where OTOP channels are
expected to function, and could be used in non-model organisms. From a medical
standpoint, most OTOP channels remain understudied in the context of normal
physiology as well as human disease, hence, it is hard to say if inhibitors or activators
would be most important for future OTOP drug discovery.

*2) The authors should clearly state which functional conformation of zfOTOP1 was used*
*for virtual screening and discuss how this choice might affect the initial hits and*
*subsequent analysis. From a functional or medical standpoint, which conformational*
*state would be most relevant to target?*

We agree that this is an important point and have edited our Results section to
include this information. (lines 111 -112, 118 - 120)

In this study, the original protein target used is a cryo-EM structure of apo
zfOTOP1 in a lipid nanodisc (PDB: 6NF4), determined at pH 8. As OTOP1 channels do
not conduct current at pH 8, the conformational state of zfOTOP1_apo likely
corresponds to a closed state of the channel. Thus, if our screen identified gating
modulators, as we believe it did, it makes sense that they would be inhibitors (binding to
and stabilizing the closed state). However, the screen could have easily identified pore
blockers that bind to and occlude the outer vestibule of the channels, although we do
not believe it identified such molecules.

*3) The authors should clarify whether the two zfOTOP1 monomers influence each*
*other's function. This might be known from earlier work, but the manuscript does not*
*currently make it clear. It would be helpful to specify whether this interaction is relevant*
*to the in silico analysis or the functional assays.*

We thank the reviewer for suggesting this. Initially, in 2019, the study by
Saotome, K. et al., *NSMB* (2019) showed that mutant zebrafish OTOP1 whose
dimerization interface were disrupted by single mutations likely don't traffic properly to
the plasma membrane and thus are non-functional.

Since then, to the best of our knowledge, only one recent study by Gan, N. et al.,
*Nat Comm* (2024), suggested that two Otopetrin monomers affect each other's function.
They showed that deletion of residues 1-57 from *Caenorhabditis elegans* OTOP8
(CeOTOP8) results in a functional channel, whereas the wild-type (WT) channel is non-
functional, and that this region interacts with the non-self subunit. However, this
remains an isolated case, as this region is not present on other OTOP channels that
have been studied and thus whether there are interactions between the two subunits
that could affect channel activity (e.g. by affecting gating) remains unclear, but is
certainly an interesting possibility.

We have addressed this point in the results section of our manuscript (lines 103-
107).

*4) Were any of the datasets processed extensively in C1 symmetry? If so, what insights*
*did that provide? Also, did the authors explore symmetry expansion to classify*
*protomers in potentially different conformations, e.g., with distinct ligand occupancies?*

We shared the same concerns as the reviewer, and we have checked whether
C1 symmetry refinement of the final particles produce a significantly different map and
we found minimal differences between ligand-attributed density between C1 and C2
refinement maps.

In addition, during our data processing, we attempted C2 symmetry expansion of
the particle sets and subsequent local refinements in C1 symmetry at various points, but
the resulting maps were always of lower quality and we never observed any significant
changes in ligand density in any of the pockets. We believe these were due to a
monomer of OTOP1 being too small and allowing for less accurate alignments of
particles than the whole dimer, especially considering that OTOP particles do not have
any extracellular domains to act as fiducial markers during alignment of particles.
Hence, we did not observe different ligand occupancies in different protomers of the
same dimer. Admittedly, we have not performed full-pipeline data processing keeping to
only C1 symmetry, but we do not expect to find major differences.

*5) The authors are encouraged to emphasize sequence similarities and identity among*
*the resolved OTOP structures, particularly between zfOTOP1 and mammalian*
*orthologs. This would help contextualize the relevance of using zfOTOP1 for virtual*
*screening, especially in light of Fig. S10, which could be referenced earlier in the results*
*section.*

We thank the reviewer for this suggestion. We have added a supplementary
figure (Supplementary Fig. 3) showing the sequence similarities of the OTOP1 orthologs
discussed in the manuscript and highlighted residues in the pockets we considered. We
have also added a few sentences to reference this figure (lines 193 - 195) and Fig.
S10A (now Supplementary Fig. 6B), earlier in the results section where relevant.

*6) Line 136: The term “visual inspection” is vague. Could the authors clarify what criteria*
*were used during this process? Were there specific features being assessed? For*
*example, were particular structural features, docking poses, or scoring thresholds taken*
*into account?*

We thank the reviewer for this question. We had mentioned this in our Methods
section (lines 602 – 605), that during visual inspection of the docked poses of
molecules, one considers the following: ideal or potential specific interactions between
the protein and ligand that were not immediately apparent from the docking pose,
proper structural conformations of ligands (e.g. checking for twisted regions where they
should be planar), consideration of desolvation of both the ligand and binding pocket
and whether the protein-ligand interactions could compensate for the desolvation cost,
the presence of desirable or unwanted functional groups in molecules, uniqueness of
chemical structure (compared to other molecules that were already selected) and
availability of stock for purchase from ChemBridge. Since docking scores were used as
a threshold at an earlier stage during data processing, we do not further consider
docking scores during visual inspection, as all ligands considered for visual inspection
are expected to have reasonably good docking scores. Admittedly, these criteria are
subjective and ultimately depend on one’s chemical intuition and rationale. We note that
the results of visual inspection were cross-checked by two of the authors in the
manuscript (BB and SF).

We have now added a brief description of visual inspection to the Results section
as well. (lines 129 - 131)

*7) The similarity between the identified scaffold and that of tricyclic antidepressants*
*(TCAs) is noteworthy. Are TCAs or structurally related compounds known to modulate*
*sour taste in humans? A brief reflection on this point would add an intriguing*
*pharmacological dimension to the findings.*

This is an interesting avenue of exploration raised by the reviewer. We have not
previously considered following up with or testing more TCA-like compounds during the
study.

It was described in two studies, by Schiffman, S.S. et al., *Physiol. Behav.* (1999)
and Schiffman, S.S. et al. *Pharmacol. Biochem. Behav.* (2000), that five TCAs broadly
reduce taste reception, including sour taste. However, this effect seems to be non-
specific among the different taste senses found to be affected, being salty, sweet, and
sour. Beyond this, to the best of our knowledge, TCAs have not been suggested to alter
sour taste reception or any known proton channels specifically.

We have only tested three dibenzazepine-based drugs, which fall under TCAs, in
our second round of screening, and none affected zfOTOP1. But given the structural
similarity of their tricyclic cores to the inhibitors we found shown in Supplementary
Figure 2E, it would not be surprising if some TCAs inhibited OTOP channels through the
intrasubunit sites. However, as TCAs are also known to non-specifically inhibit various
ion channels, including sodium, potassium, L-type calcium, and ligand-gated ion
channels, any inhibition of OTOPs by TCAs could be due to non-specific interactions,
which would need to be distinguished from its specific interactions, if any.

We have added a succinct version of the points above into our discussion
section. (lines 451 - 459)

*8) While the authors hypothesize that the binding of compounds such as C11 and C2.2*
*may not involve interactions with titratable residues, could it be that these ligands*
*stabilize a conformation of zfOTOP1 that indirectly affects titratable residues? The*
*authors are encouraged to reflect on this possibility, as it may influence the*
*interpretation of their binding mechanism.*

We thank the reviewer for raising this point. We agree that certain titratable
residues could be indirectly affected as a result of inhibitor binding. Here, we expect that
any such indirect effect still be mediated through structural changes that occur upon
inhibitor binding, through shifting of the backbone or sidechain rotamers of titratable
residues.

In our cryo-EM structures, the structural changes are highly localized to the
intrasubunit interface area. However, in this region, we see no clear differences
between any titratable residues, namely E267, H574 and D382, between the apo
structure and inhibitor bound structures. One difference we see is that TM9, which
contains D382 (not shown in figures), is shifted in our inhibitor-bound structures
(Supplementary Figure 7), such that it places the side chain of D382 closer to that of
H574. These residues form a hydrogen bond interaction (even in the apo structure), are
well-conserved among Otopetrins, and important for channel function and gating, as
H574 is essential to channel function (Saotome, K. et al., *NSMB* (2019)) and D382
(D388 in mouse OTOP1) was shown to influence the rate of channel desensitization
(Chen, Q. et al. *eLife* (2019)). Perhaps this difference, resulting from the binding of an

inhibitor, influences the interaction between these two residues, contributing to the
inhibition of zfOTOP1.

*9) The manuscript uses CHS and CHOL somewhat interchangeably in describing bound*
*lipid molecules. Given the similar density features of these sterols in cryo-EM maps,*
*how were they distinguished in the models? If the presence of CHS and/or CHOL was*
*confirmed experimentally, for instance by mass spectrometry, this should be explicitly*
*mentioned in the text.*

We admit that we have used cholesterol and CHS somewhat interchangeably
throughout the manuscript. Regarding the biochemistry, our protein expression and
purification follow the same protocols as the previous study by Saotome, K. et al, *NSMB*
(2019), which presented the structure of apo zfOTOP1, where CHS is added to the
detergent buffers. Thus, we expect that the majority of cholesterol-like density we find
correspond to CHS. However, it is possible that native cholesterol molecules bound to
zfOTOP1 are carried over through the entire purification process. So, it remains
possible that any cholesterol-like density could correspond to cholesterol as well. We
have not performed any mass spectrometry or other biophysical assays to determine
whether the bound lipids are cholesterol or CHS, although we expect a mix of both.

Here, our basis for modelling these molecules followed the modelling of CHS or
cholesterol in the apo structure of zfOTOP1 (PDB: 6NF4), where six CHS molecules
were modelled in the central cavity of zfOTOP1, including one in the central site, and
one cholesterol molecule each were modelled into the intrasubunit outer sites. For our
structures, we attempted to model in the same molecule as the apo structure of
zfOTOP1 wherever possible, and for any extra cholesterol-like density identified, we
modelled in cholesterol, as it usually fit better into the density.

We have added a short statement clarifying how we modelled in cholesterol or
CHS into our cryo-EM maps into our Methods section. (lines 864 - 866)

References:

- - Chen, Q., Zeng, W., She, J., Bai, X.-C. & Jiang, Y. Structural and functional
characterization of an otopetrin family proton channel. *Elife* **8**, (2019).
- - Gan, N., Zeng, W., Han, Y., Chen, Q. & Jiang, Y. Structural mechanism of proton
conduction in otopetrin proton channel. *Nat Commun* **15**, 7250 (2024).
- - Mi, T., Mack, J. O., Lee, C. M. & Zhang, Y. V. Molecular and cellular basis of acid
taste sensation in *Drosophila*. *Nat Commun* **12**, 3730 (2021).
- - Saotome, K. *et al.* Structures of the otopetrin proton channels Otop1 and Otop3. *Nat*
*Struct Mol Biol* **26**, 518–525 (2019).
- - Schiffman, S. S., Zervakis, J., Suggs, M. S., Shaio, E. & Sattely–Miller, E. A. Effect
of Medications on Taste: Example of Amitriptyline HCl. *Physiology & Behavior* **66**,
183–191 (1999).
- - Schiffman, S. S., Zervakis, J., Suggs, M. S., Budd, K. C. & Iuga, L. Effect of Tricyclic
Antidepressants on Taste Responses in Humans and Gerbils. *Pharmacology*
*Biochemistry and Behavior* **65**, 599–609 (2000).
- - Teng, B. *et al.* Structural motifs for subtype-specific pH-sensitive gating of vertebrate
otopetrin proton channels. *eLife* **11**, e77946 (2022).

REVIEWERS' COMMENTS

Reviewer #1 (Remarks to the Author):

The authors have adequately addressed all of my concerns. I recommend this paper for
publication.

We thank the reviewer for taking the time and effort to review our manuscript.

Reviewer #2 (Remarks to the Author):

*The authors have addressed most of my concerns.*

*For their response to my comment #1, rather than performing blind docking, would the*
*authors consider conducting new docking studies using the current molecules specifically*
*on the binding pockets identified in the Cryo-EM structures? This approach would allow for*
*a focused evaluation of the experimentally determined sites. It is particularly valuable for*
*the binding pockets that in the case where two C2.36 molecules were assigned to the*
*central site, where the cryo-EM densities are not very clear. In their response to my*
*comment #8, the authors also mention that they are uncertain about the assignment of*
*C2.36 at that site, particularly the second copy.*

As the reviewer requested, we performed a docking of C2.2 and C2.36 into each of
the intrasubunit sites of their cryo-EM models, DrOTOP1_C2.2 and Dr_OTOP1_C2.36,
respectively. These results are now shown as Supplementary Figure 9. In brief, using
AutoDock-GPU (v1.6), which is the current standard for AutoDock, we were able to find
docked poses that closely resembled the cryo-EM modelled poses for both inhibitors. For
the outer site, we find that the most similar poses are found with RMSDs of $<2.8 \text{ \AA}$, and for
the central site, with RMSDs of $<2.0 \text{ \AA}$, for both inhibitors. This includes the second copy of
C2.36 in the central site we modelled as well. In the field, the conventional cutoff by RMSD
is $<2 \text{ \AA}$, in order to claim that experimental poses have been reproduced by docking. While
the outer site docked poses do not exactly meet this cutoff, visually comparing the poses
shows that they are still similar. (Supplementary Fig. 9A, B). Hence, we believe the
redocking was successful in indicating to us that the intrasubunit interface binding sites
are reasonable sites for small molecules to occupy, to the degree that molecular docking
can answer.

However, we note that the most similar docked poses we showed are not the top
scoring poses but fall within a cutoff of $< 1 \text{ kcal/mol}$ in docking score from the top scoring
poses, which are still considered successful dockings by AutoDock-GPU by its default
behavior. We hypothesize that this shortcoming is due to the nature of the intrasubunit
sites, which are highly hydrophobic and lipid exposed, which are traits we initially deemed
as unideal for virtual screening at the beginning of the study. This is because the current
scoring functions of AutoDock related software do not accurately consider contributions
from small molecule-lipid interactions and desolvation costs at such lipid exposed sites,

causing the scores to be inaccurate and potentially favor less-than-ideal poses as the
global minimum poses. This remains a challenge to be addressed in the field of molecular
docking, especially for membrane proteins.

We have included our results and reasoning above as brief paragraphs in the results
(lines: 360 – 370) and discussion sections (lines: 529-534), and we have summarized our
methodology in the methods section as well (lines: 684-699). Line numbers correspond to
the “All Markup” view.

*Also, since the Cryo-EM structures have shown that the initial docking sites were*
*incorrect, I believe it would be more appropriate to move the docking results currently*
*shown in Figure 1 to the supplementary information, rather than including them in the main*
*text.*

We thank the reviewer for this suggestion and agree that this could confuse readers.
We have now split Figure 1 into a new Figure 1 and Supplementary Figure 1, which removes
docked poses in the C-domain pocket from the main figures.

*Reviewer #4 (Remarks to the Author):*

*This is a compelling manuscript reporting the discovery of several new modulators of*
*OTOP1. The data is of excellent quality and well-presented. I was brought on to the review*
*in place of reviewer #3. I find that the authors have addressed the major concerns of*
*reviewer #3, and I congratulate the authors on this work.*

We thank the reviewer for taking the time and effort to review our manuscript in
place of the former reviewer.
